# HomeRobot: Open-Vocabulary Mobile Manipulation

**Sriram Yenamandra**[*1]  **Arun Ramachandran**[*1]  **Karmesh Yadav**[*1,2]  **Austin Wang**[1]
**Mukul Khanna**[1]  **Theophile Gervet**[2,3]  **Tsung-Yen Yang**[2]  **Vidhi Jain**[3]
**Alexander William Clegg**[2]  **John Turner**[2]  **Zsolt Kira**[1]  **Manolis Savva**[4]
**Angel Chang**[4]  **Devendra Singh Chaplot**[2]  **Dhruv Batra**[1,2]  **Roozbeh Mottaghi**[2]
**Yonatan Bisk**[2,3]  **Chris Paxton**[2]

[1]Georgia Tech  [2]FAIR, Meta AI  [3]Carnegie Mellon  [4]Simon Fraser
**homerobot-info@googlegroups.com**

**Abstract: HomeRobot** (*noun*): An affordable compliant robot that navigates homes and manipulates a wide range of objects in order to complete everyday tasks.

Open-Vocabulary Mobile Manipulation (OVMM) is the problem of picking *any* object in *any* unseen environment, and placing it in a commanded location. This is a foundational challenge for robots to be useful assistants in human environments, because it involves tackling sub-problems from across robotics: perception, language understanding, navigation, and manipulation are all essential to OVMM. In addition, integration of the solutions to these sub-problems poses its own substantial challenges. To drive research in this area, we introduce the HomeRobot OVMM benchmark, where an agent navigates household environments to grasp novel objects and place them on target receptacles. HomeRobot has two components: a *simulation* component, which uses a large and diverse curated object set in new, high-quality multi-room home environments; and a *real-world* component, providing a software stack for the low-cost Hello Robot Stretch to encourage replication of real-world experiments across labs. We implement both reinforcement learning and heuristic (model-based) baselines and show evidence of sim-to-real transfer of the nav and place skills. Our baselines achieve a 20% success rate in the real world; our experiments identify ways future work can improve performance. See videos on our website: https://ovmm.github.io/.

**Keywords:** Sim-to-real, benchmarking robot learning, mobile manipulation

## 1  Introduction

The aspiration to develop household robotic assistants has served as a north star for roboticists since the beginning of the field. The pursuit of this vision has spawned multiple areas of research within robotics from vision to manipulation, and has led to increasingly complex tasks and benchmarks. A useful household assistant requires creating a capable mobile manipulator that understands a wide variety of objects, how to interact with the environment, and how to intelligently explore a world with limited sensing. This has separately motivated research in diverse areas like navigation [1, 2], service robotics [3–5], language understanding [6, 7] and task and motion planning [8]. We refer to this guiding problem as *Open-Vocabulary Mobile Manipulation (OVMM):* a useful robot will be able to find and move arbitrary objects from place to place in an arbitrary home.

Prior work does not tackle mobile manipulation in large, continuous, real-world environments. Instead, it generally simplifies the setting significantly, e.g. by using discrete action spaces, limited object sets, or small, single-room environments that are easily explored. However, recent developments tying language and vision have enabled robots to generalize beyond specific categories [9–13], often through multi-modal models such as CLIP [14]. Further, comparison across methods has remained difficult and reproduction of results across labs impossible, since many aspects of the

7th Conference on Robot Learning (CoRL 2023), Atlanta, USA.

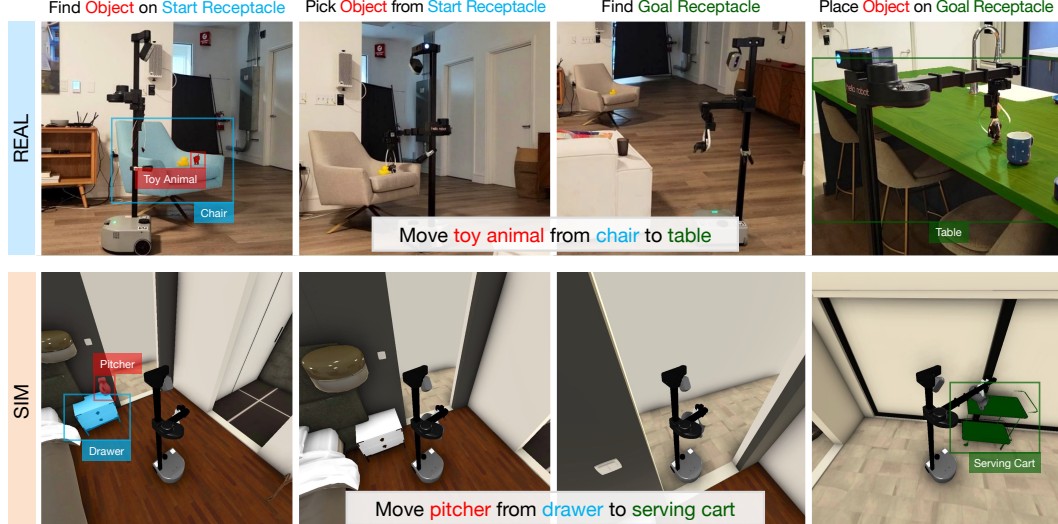

Figure 1: Open-Vocabulary Mobile Manipulation requires agents to search for a previously unseen object at a particular location, and move it to the correct receptacle.

settings (environments, and robots) have not been standardized. This is especially important now, as a new wave of research projects have begun to show promising results in complex, open-vocabulary navigation [9, 15, 11, 12, 16] and manipulation [17, 10, 18] – again on a wide range of robots and settings, and still limited to single-room environments. Clearly, now is the time when we need a common platform and benchmarks to drive the field forward.

In this work, we define Open-Vocabulary Mobile Manipulation as a key task for in-home robotics and provide benchmarks and infrastructure, both in simulation and the real world, to build and evaluate full-stack integrated mobile manipulation systems, in a wide variety of human-centric environments, with open object sets. Our benchmark will further reproducible research in this setting, and the fact that we support arbitrary objects will enable the results to be deployed in a variety of real-world environments.

**OVMM:** We propose the first reproducible mobile-manipulation benchmark for the real world, with an associated simulation component. In simulation, we use a dataset of 200 human-authored interactive 3D scenes [19] instantiated in the AI Habitat simulator [20, 21] to create a large number of challenging, multi-room OVMM problems with a wide variety of objects curated from a variety of sources. Some of these objects' categories have been seen during training; others have not. In the real world, we create an equivalent benchmark, also with a mix of seen and unseen object categories, in a controlled apartment environment. We use the Hello Robot Stretch [22]: an affordable and compliant platform for household and social robotics that is already in use at over 40 universities and industry research labs. Fig. 1 shows instantiations of our OVMM task in both the real-world benchmark and in simulation. We have a controlled real-world test environment, and plan to run the real-world benchmark yearly to assess progress on this challenging problem. Real-world benchmarking will be run as a part of the NeurIPS 2023 HomeRobot OVMM competition [23].

**HomeRobot:** We also propose HomeRobot,[1] a software framework to facilitate extensive benchmarking in both simulated and physical environments. It comprises identical APIs that are implemented across both settings, enabling researchers to conduct experiments that can be replicated in both simulated and real-world environments. Table 1 compares HomeRobot OVMM to the literature. Notably, HomeRobot provides a robotics stack for the Hello Robot Stretch which supports a range of capabilities in both simulation and the real world, and is not restricted to just the OVMM task. Our library also supports a number of sub-tasks, including manipulation learning [24], continuous learning [25], navigation [26], and object-goal navigation [2].

---

[1] https://github.com/facebookresearch/home-robot

| | | Object | | Continuous | | Robotics | Open | |
|---|---|---|---|---|---|---|---|---|
| | Scenes | Cats | Inst. | Actions | Sim2Real | Stack | Licensing | Manipulation |
| Room Rearrangement [28] | 120 | 118 | 118 | ✗ | ✗ | ✗ | ✔ | ✗ |
| Habitat ObjectNav Challenge [29] | 216 | 6 | 7,599 | ✔ | ✗ | ✗ | ✔ | ✗ |
| TDW-Transport [30] | 15 | 50 | 112 | ✗ | ✗ | ✗ | ✓ | ✓ |
| VirtualHome [31] | 6 | 308 | 1,066 | ✗ | ✗ | ✗ | ✔ | ✓ |
| ALFRED [6] | 120 | 84 | 84 | ✗ | ✗ | ✗ | ✔ | ✓ |
| Habitat 2.0 HAB [21] | 105 | 20 | 20 | ✔ | ✗ | ✗ | ✔ | ✔ |
| ProcTHOR [32] | 10,000 | 108 | 1,633 | ✗ | ✗ | ✗ | ✔ | ✔ |
| RoboTHOR [33] | 75 | 43 | 731 | ✗ | ✔ | ✗ | ✔ | ✗ |
| Behavior-1K [34] | 50 | 1,265 | 5,215 | ✔ | ✔ | ✗ | ✗ | ✓ |
| ManiSkill-2 [35] | 1 | 2,000 | 2,000 | ✔ | ✓ | ✗ | ✓ | ✔ |
| 🤖 OVMM + HomeRobot | 200 | 150 | 7,892 | ✔ | ✔ | ✔ | ✔ | ✔ |

Table 1: Comparisons of our proposed benchmark with prior work. We provide a large number of environments and unique objects, focusing on manipulable objects, with a continuous action space. Uniquely, we also provide a multi-purpose, real-world robotics stack, with demonstrated sim-to-real capabilities, allowing others to reproduce and deploy their own solutions. Additional nuances in footnote[3]. ✓Partial availability ✗Not available ✔Capability available

In this paper, we use HomeRobot to compare two families of approaches: a *heuristic* solution, using a motion planner shown to work for real-world object search [2], and a *reinforcement learning* (RL) solution, which learns how to navigate to objects given depth and predicted object segmentation. We use the open-vocabulary object detector DETIC [27] to provide object segmentation for both the heuristic and RL policies. We observe that while the RL methods moved to the object more efficiently if an object was visible, the heuristic planner was better at long-horizon exploration. We also see a substantial drop in performance when switching from ground-truth segmentation to DETIC segmentation. This highlights the importance of the HomeRobot OVMM challenge, as only through viewing the problem holistically - integrating perception, planning, and action - can we build general-purpose home assistants.

To summarize, in this paper, we define Open-Vocabulary Mobile Manipulation as a new, crucial task for the robotics community in Sec. 3. We provide a new simulation environment, with multiple, multi-room interactive environments and a wide range of objects. We implement a robotics library called HomeRobot which provides baseline policies implementing this in both the simulation and the real world. We describe a real-world benchmark in a controlled environment, and show how current baselines perform in simulation and in the real world under different conditions. We plan to initially run this real-world benchmark as a Neurips 2023 competition [23].

## 2   Related Work

We discuss work related to challenges and reproducibility of robotics research in more detail, but continue the discussion of datasets and simulators in Appendix A.

**Challenges.** There have been several challenges aiming to benchmark robotic systems at different tasks. These challenges provided a great testbed for ranking different systems. However, in most of the challenges (e.g., [36–39, 3]), the participants create their own robotic platform making a fair comparison of the algorithms difficult. There are also challenges where the organizers provide the robotic platform to the participants (e.g., [40]). However, changing the task during the periodic evaluations made it difficult to track progress over time. Our aim is to have a real world benchmark using a standard hardware that is sustainable at least for a few years.

**Reproducibility of robotics research.** Standardized robotics benchmarks have been pursued for a long time, often by open-sourcing robot designs or introducing low-cost robots [41–49]. However, the environments in which these robots are used vary dramatically, leading to evaluation of components (e.g., object navigation, SLAM) in isolation, instead of as components of a larger system that

---

[3]ALFRED uses object masks for interaction. ObjectNav uses scans, not full object meshes. ProcThor scenes are procedurally generated, this has the benefit that the potential number of environments is unbounded.

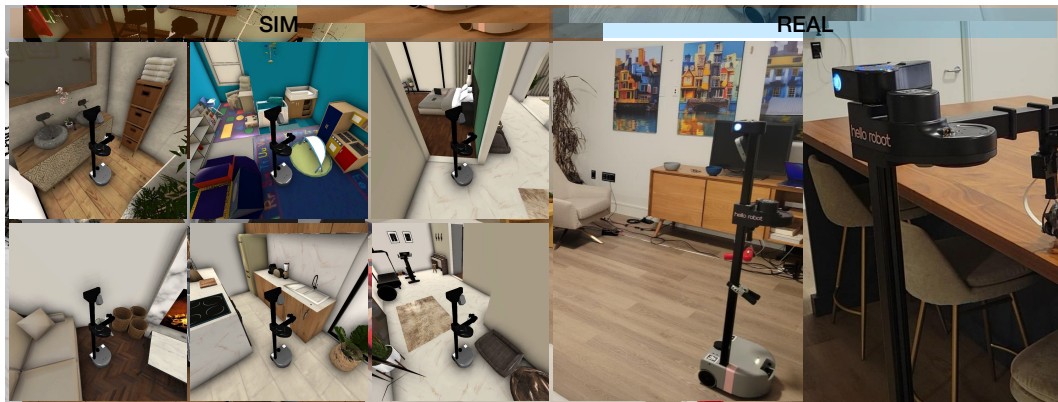

Figure 2: A low-cost home robot performing tasks in both a simulated and a real-world environment. We provide both (1) challenging simulated tasks, wherein a mobile manipulator robot must find and grasp multiple seen and unseen objects, and (2) a corresponding real-world robotics stack to allow others to reproduce this research and evaluation to produce useful home robot assistants.

may not benefit from those changes. The HomeRobot stack enables end-to-end benchmarking of individual components by providing a full robotics stack, with multiple implementations of different sub-modules. The simplicity helps move beyond standardized sets of objects (e.g., [50–52]) to a common set of robots, objects, and environments. Ours is the only benchmark to provide a *broadly capable* robotics stack for implementing and sharing robotics code; this is similar to projects like PyRobot [53], which doesn't also provide a strong simulation benchmark.

**Real World Benchmarks.** RoboTHOR [33] provides a common set of scenes and objects for benchmarking navigation. RB2 [54] ranks different manipulation algorithms in a local setting. TOTO [55] takes a step further by providing a training dataset and running the experiments for the users. However, training and testing happen in the same environments and are limited to tabletop manipulation. Finally, the NIST Task Board [56] is a successful challenge for fine-grained manipulation skills [57], also limited to a tabletop context. Kadian et al. [58] propose the Habitat-PyRobot bridge (HaPy) to allow real-world testing on the locobot robot; their framework is limited to navigation, and doesn't provide a generally-useful robotics stack with visualizations, debugging, motion planners, tooling, etc.

## 3 Open-Vocabulary Mobile Manipulation

Formally, our task is set up as instructions of the form: "Move (`object`) from the (`start_receptacle`) to the (`goal_receptacle`)." The `object` is a small and manipulable household object (e.g., a cup, stuffed toy, or box). By contrast, `start_receptacle` and `goal_receptacle` are large pieces of furniture, which have surfaces upon which objects can be placed. The robot is placed in an unknown single-floor home environment - such as an apartment - and must, given the language names of `start_receptacle`, `object`, and `goal_receptacle`, pick up an `object` that is known to be on a `start_receptacle` and move it to any valid `goal_receptacle`. `start_receptacle` is always available, to help agents know where to look for the `object`.

The agent is successful if the specified `object` is indeed moved from a `start_receptacle` on which it began the episode, to any valid `goal_receptacle`. We give partial credit for each step the robot accomplishes: finding the `start_receptacle` with the `object`, picking up the `object`, finding the `goal_receptacle`, and placing the `object` on the `goal_receptacle`. There can be multiple valid objects that satisfy each query.

Crucially, we need and develop both (1) a simulation version of the Open-Vocabulary Mobile Manipulation problem, for reproducibility, training, and fast iteration, and (2) a real-robot stack with a corresponding real-world benchmark. We compare the two in Fig. 2. Our simulated environments allow for varied, long-horizon task experimentation; our real-world HomeRobot stack allows for

experimenting with real data, and we design a set of real-world tests to evaluate the performance of our learned and heuristic baselines.

**The Robot.** We use the Hello Robot Stretch [22] with DexWrist as the mobile manipulation platform, because it (1) is *relatively* affordable at $25,000 USD, (2) offers 6 DoF manipulation, and (3) is human safe and human-sized, making it safe to test in labs [24, 11] and homes [2], and can reach most places a human would expect a robot to go. For a breakdown of hardware choices, see Sec. H.1.

**Objects.** These are split into *seen* vs. *unseen categories* and *instances*. In particular, at test time we look at unseen instances of seen or unseen categories; i.e. no seen manipulable object from training appears during evaluation. Agents must pick and place any requested object.

**Receptacles.** We include common household receptacles (e.g. tables, chairs, sofas) in our dataset; unlike with manipulable objects, all possible receptacle categories are seen during training.

**Scenes.** We have both a simulated scene dataset and a fixed set of real-world scenes with specific furniture arrangements and objects. In both simulated and real scenes, we use a mixture of objects from *previously-seen* categories, and objects from *unseen* categories as the goal `object` for our Open-Vocabulary Mobile Manipulation task. We hold out *validation* and *test* scenes, which do not appear in the training data; while some receptacles may re-appear, they will be at previously unseen locations, and target object instances will be unseen.

**Scoring.** We compute success for each stage: finding `object` on `start_receptacle`, successfully picking up `object`, finding `goal_receptacle`, and placing `object` on the goal. Overall success is true if all four stages were accomplished. We compute *partial success* as a tie-breaker, in which agents receive 1 point for each successive stage accomplished, normalized by the number of stages. More details in Appendix C.

## 3.1 Simulation Dataset

The Habitat Synthetic Scenes Dataset (HSSD) [19] consists of 200+ human-authored 3D home scenes containing over 18k 3D models of real-world objects. Like most real houses, these scenes are cluttered with furniture and other objects placed into realistic architectural layouts, making navigation and manipulation similarly difficult to the real world. We used a subset of HSSD [19] consisting of 60 scenes for which additional metadata and simulation structures were authored to support rearrangement [4]. For our experiments, these are divided into train, validation, and test splits of 38, 12, and 10 scenes each, following the splits in the original HSSD paper [19].

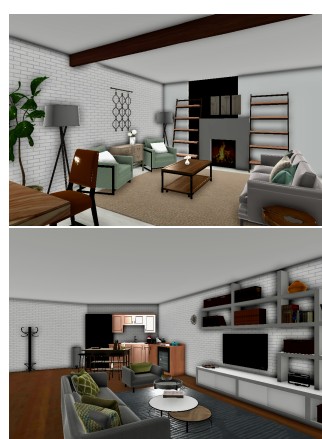

**Objects and Receptacles.** We aggregate objects from AI2-Thor [59], Amazon-Berkeley Objects [60], Google Scanned Objects [61] and the HSSD [19] dataset to create a large and diverse dataset of real-world robot problems. In total, we annotated 2,535 objects from 129 total categories. We identified 21 different categories of receptacles which appear in the HSSD dataset [19].

Figure 3: HSSD scenes.

We construct our final set of furniture receptacle objects by first automatically labeling stable areas on top of receptacles, then manually refining and processing these in order to remove invalid or inaccessible receptacles. In addition, collision proxy meshes were automatically generated and in many cases manually corrected to support physically accurate procedural placement of object arrangements.

|       | SC, SI | SC, UI | UC, UI | Total |
|-------|--------|--------|--------|-------|
| Cats  | 85     | 64     | 44     | 129   |
| Insts | 1,363  | 748    | 424    | 2,535 |

Table 2: # of objects in the sim for each split of (S)een and (U)nseen (I)nstance and (C)ategory.

---

[4]All 200+ scenes with rearrangement support will be released soon.

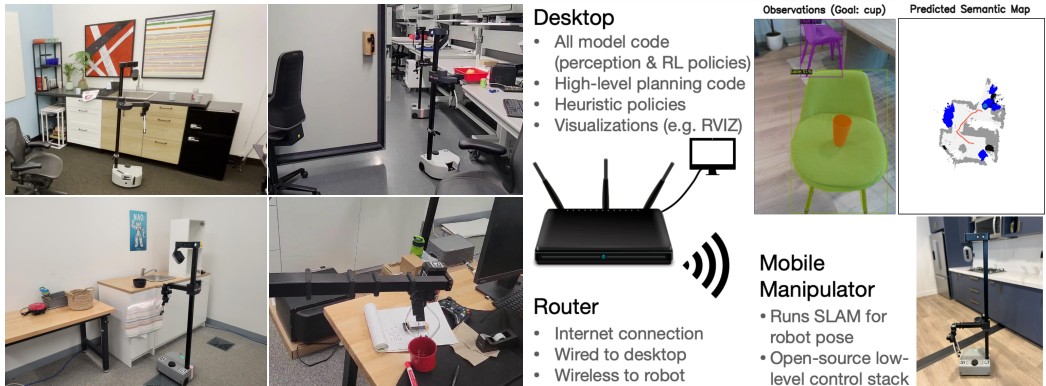

Figure 4: 🤖HomeRobot is a simple, easy-to-set-up library that works in multiple environments and requires only relatively affordable hardware. Computationally intensive operations are performed on a desktop PC with a GPU, and a dedicated consumer-grade router provides a network interface to a robot running low-level control and SLAM.

**Episode Generation.** We generate episodes consisting of varying object arrangements and particular values for `object`, `start_receptacle`, and `goal_receptacle`, which allow our agent to successfully move about and interact with the world. In the case of Open-Vocabulary Mobile Manipulation, this task is particularly challenging because we have to place objects in locations that are *navigable*, meaning that the robot can get to them, *reachable*, meaning its arm can make it to these locations, and from which we can navigate to a *navigable*, *reachable* goal receptacle. For full episode generation details see App. D.2.

**Training and Validation Split.** Training episodes consist of objects from the large pool of seen instances of seen categories *(SC,SI)*. In contrast, we use unseen instances of seen object categories *(SC,UI)* and unseen instances of unseen categories *(UC,UI)* for validation and test episodes. Two-thirds of the categories were randomly designated as seen, and two-thirds of the objects in the seen category were randomly marked as seen instances. Splits are in Table 2 and the distribution of objects across categories is in App. Fig. 6.

## 3.2 Real World Benchmark

Real-world experiments are performed in a controlled 3-room apartment environment, with a sofa, kitchen table, counter with bar, and TV stand, among other features. We documented the positioning of various objects and the robot start position, in order to ensure reproducibility across trials. Images of various layouts of the test apartment are included in Fig. 2, and task execution is shown in Fig. 16.

During real-world testing, we selected object instances that did not appear in simulation training, split between classes that did and did not appear. We used eight different categories: five seen (*Cup*, *Bowl*, *Stuffed Toy*, *Medicine Bottle*, and *Toy Animal*), and three unseen (*Rubik's cube*, *Toy Drill*, and *Lemon*). We performed 20 experiments for each of our two different baselines and with seven different receptacle classes: *Cabinet*, *Chair*, *Couch*, *Counter*, *Sink*, *Stool*, *Table*.

## 4  The 🤖HomeRobot Library

To facilitate research on these challenging problems, we open-source the HomeRobot library, which implements navigation and manipulation capabilities supporting Hello Robot's Stretch [22]. In our setup, it is assumed that users have access to a mobile manipulator and an NVIDIA GPU-powered workstation. The mobile manipulator runs the low-level controller and the localization module, while the desktop runs the high-level perception and planning stack(Fig. 4). The robot and desktop are connected using an off-the-shelf router[5]. The key features of our stack include:

---

[5]Our experiments used a NetGear Nighthawk router.

**Transferability:** Unified state and action spaces between simulation & real-world settings for each task, providing an easy way to control a robot with either high-level action spaces (e.g., pre-made grasping policies) or low-level continuous joint control.

**Modularity:** Perception and action components to support high-level states (e.g. semantic maps, segmented point clouds) and high-level actions (e.g. go to goal position, pick up target object).

**Baseline Agents:** Policies that use these capabilities to provide basic functionality for OVMM.

### 4.1 Baseline Agent Implementation

Crucially, we provide baselines and tools that enable researchers to effectively explore the Open-Vocabulary Mobile Manipulation task. We include two types of baselines in HomeRobot: a heuristic baseline, using motion planning [2] and simple rules for manipulation; and a reinforcement learning baseline. In addition, we have implemented example projects from several recently released papers, testing different capabilities such as object-goal navigation [1, 2], skill learning [24], continual learning [25], and image instance navigation [26]. We implement a high-level policy called `OVMMAgent` which calls a sequence of skills one after the other. These skills are:

**FindObj/FindRec:** Locate an `object` on a `start_receptacle`; or find a `goal_receptacle`.

**Gaze:** Move close enough to an `object` to grasp it, and orient head to get a good view of the object. The goal of the gaze action is to improve the success rate of grasping.

**Pick:** Pick up the `object`. We provide a high-level action for this, since we do not simulate the gripper interaction in Habitat. However, our library is compatible with a range of learned grasping skills and supports learning policies for grasping.

**Place:** Move to a location in the environment and place the `object` on top of the `goal_receptacle`.

Specifically, `OVMMAgent` is a state-machine that calls **FindObj**, **Gaze**, **Pick**, **FindRec**, and **Place** in that order, where **Pick** is a grasping policy provided by the robot library in the real world. The other skills are created using the approaches given below:

**Heuristic.** We implement a version using only off-the-shelf learned models and heuristics, noting that previous work in mobile manipulation has used these models to great effect (e.g. [62]). Here, DETIC [63] provides masks for an open-vocabulary set of objects as appropriate for each skill. The `start_receptacle`, `object`,`goal_receptacle` for each episode is given. Fig. 16 shows an example of the heuristic navigation and place policy being executed in the real world (App. E).

**RL.** We train the four skills in our modified version of Habitat [21] as policies which predict actions given depth, ground truth semantic segmentation and prioprecptive sensors (i.e. joints, gripper state), using DDPPO [64]. While RGB is available in our simulation, our baseline policies do not directly utilize it; instead, they rely on predicted segmentation from Detic [27] at test time.

## 5 Results

We first evaluate the two baselines in our simulated benchmark, followed by evaluation in a real-world, held-out test apartment. These results highlight the significance of OVMM as a challenging new benchmark, encompassing numerous essential challenges that arise when deploying robots in real-world environments.

We break down the results by sub-task in addition to reporting the overall performance in Tables 3 and 4. The columns **FindObj**, **Pick** and **FindRec** refer to the first 3 phases of the task mentioned in the scoring section (Sec. 3), and succeeding in the final Place phase leads to a successful episode.

**Simulation.** We evaluate the baselines on held-out scenes, with objects from unseen instances of seen classes, and unseen instances of *unseen* classes, as described in Sec. 3.1. We show results with two different perception systems: **Ground Truth** segmentation, where we use the segmentation input directly from the simulator, and **DETIC** segmentation [27], where the RGB images from the simulator are passed through DETIC, an open-vocabulary object detector.

| Simulation Results | Skill | | | Partial Success Rates | | | Overall Success Rate | Partial Success Metric |
|---|---|---|---|---|---|---|---|---|
| Perception | Navigation | Gaze | Place | FindObj | Pick | FindRec | | |
| Ground Truth | Heuristic | None | Heuristic | 54.1 | 48.5 | 31.5 | 5.1 | 34.8 |
| | Heuristic | RL | RL | 56.5 | 51.5 | 42.3 | 13.2 | 40.9 |
| | RL | None | Heuristic | 65.4 | 54.8 | 43.7 | 7.3 | 42.8 |
| | RL | RL | RL | 66.6 | 61.1 | 50.9 | 14.8 | 48.3 |
| DETIC [27] | Heuristic | None | Heuristic | 28.7 | 15.2 | 5.3 | 0.4 | 12.4 |
| | Heuristic | RL | RL | 29.4 | 13.2 | 5.8 | 0.5 | 12.2 |
| | RL | None | Heuristic | 21.9 | 11.5 | 6.0 | 0.6 | 10.0 |
| | RL | RL | RL | 21.7 | 10.2 | 6.2 | 0.4 | 9.6 |

Table 3: Partial and overall success rate (SR) (in %) for different combinations of skills and perception systems. The partial SR for each skill is dependent on the previous skill's SR. The partial SR for the place skill is the same as the overall SR. The partial success metric is calculated by averaging the 4 partial SRs. One of the main causes of failures for our baseline systems was perception; ground-truth perception is notably better. Both RL and heuristic skills struggled to navigate tightly constrained multi-room environments and successfully place objects.

We report results on HomeRobot OVMMin Table 3. RL policies outperformed heuristic methods for both navigation and placement tasks. However, all policies declined in performance when using DETIC instead of ground truth segmentation. Heuristic policies exhibited less degradation in performance compared to RL policies: when using DETIC, the heuristic FindObj policy even outperforms RL. We attribute this to the heuristic policy's ability to incorporate noisy predictions by constructing a 2D semantic map, which helps handle small objects that are prone to misclassification. Furthermore, using the learned gaze policy led to improved pick performance, except when used in combination with the Heuristic nav with DETIC perception. Example simulation trajectories can be found in Appendix Figure 18, and comparisons of seen versus unseen categories in Appendix G.2.

**Real World.** Finally, we conducted a series of experiments in a real-world held-out apartment setting. We performed a total of 20 episodes, utilizing a combination of seen and unseen object classes as our target objects. The results of these experiments are presented in Table 4. RL performed slightly better than the Heuristic baseline, successfully completing 1 extra episode and achieving a success rate of 20%. This difference primarily stemmed from the pick and place sub-tasks. In the pick task, the RL Gaze skill plays a crucial role in achieving better alignment between the agent and the target object, which leads to more successful grasping. Similarly, the RL place skill demonstrated more precision, ensuring that the object stayed closer to the surface of the receptacle.

Both simulation and real-world results show the baselines are promising, but insufficient, for Open-Vocabulary Mobile Manipulation. DE-TIC [27] caused many failures due to misclassification, both in simulation and the real world. Further, RL navigation was on par or better than heuristic policies in both sim and real. Although our RL place policy performed better in sim than heuristic place, it needs further improvement in the real world. Gaining the advantages of web-scale pretrained vision-language models like DETIC, but tuned to our agents may be crucial for improving performance.

| Real World | FindObj | Pick | FindRec | Overall Success |
|---|---|---|---|---|
| Heuristic Only | 70 | 35 | 30 | 15 |
| RL Only | 70 | 45 | 30 | 20 |

Table 4: Success Rate (in %) for heuristic and RL baselines in the real world OVMM task. In both cases, the grasping action is executed as described in Sec. 4; but initial conditions of the robot such as its position relative to the object or to other obstacles may cause various failures.

## 6 Conclusions and Future Work

We proposed a combined simulation and real-world benchmark to enable progress on the important problem of Open-Vocabulary Mobile Manipulation. We ran extensive experiments showing promising simulation and real-world results from two baselines: a heuristic baseline based on a state-of-the-art motion planner [2] and a reinforcement learning baseline trained with DDPPO [64]. In the future, we hope to improve the complexity of the problem space, adding more complex natural language and multi-step commands, and provide end-to-end baselines instead of modular policies.

# 7 Acknowledgements

We would like to thank Andrew Szot for help with Habitat policy training, Santhosh Kumar Ramakrishnan with help on Stretch Object navigation in simulation and on the real robot, and Eric Undersander for help with improving Habitat rendering. Priyam Parashar, Xiaohan Zhang, and Jay Vakil helped with testing on Stretch and real-world scene setup.

We would also like to thank the whole Hello Robot team, but especially Binit Shah and Blaine Matulevich for their help with the robots, and Aaron Edsinger and Charlie Kemp for helpful discussions.

The Georgia Tech effort was supported in part by ONR YIP and ARO PECASE. The views and conclusions contained herein are those of the authors and should not be interpreted as necessarily representing the official policies or endorsements, either expressed or implied, of the U.S. Government, or any sponsor.

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

# Appendix

## Table of Contents

## A   Extended Related Work

It is difficult to do justice to the rich embodied AI, natural language, computer vision, machine learning, and robotics communities that have addressed aspects of the work presented here. The following extends some of the discussion from the main manuscript about important advances that the community has made.

Benchmarks have helped the community focus their efforts and fairly compare system performance. For example, the YCB objects [50] allowed for direct comparison of results across manipulators and models. While benchmarks and leaderboards are comparatively rare in robotics [49, 56, 33, 54, 63, 3, 39], they have been hugely influential in machine learning (e.g. ImageNet [65], GLUE [66], and various language benchmarks [67–70], COCO [71], and SQuAD [72]). In robotics, competitions such as RoboCup@Home [3], the Amazon Picking Challenge [39], and the NIST task board [56] are prevalent and influential as an alternative, but generally systems aren't reproducible across teams.

**Datasets.** In addition to the environments referenced in Table 1, offline datasets including robot interactions with scenes have been used widely to train models. These datasets are typically obtained using robots alone (e.g., [73, 74]), by teleoperation (e.g., [75, 76]) or human-robot demonstration (e.g., [77]). Previous work such as [78] aim to collect large-scale datasets while works such as [79] consider scaling across multiple embodiments. [80] take a step further by collecting robot data in unstructured environments. Unlike these works, we do not limit our users to a specific dataset. Instead, we provide a simulator with various scenes that can generate large-scale consistent data for training. Also, note that we test the models in unseen environments, while most of the mentioned works use the same environment for training and testing.

**Simulation benchmarks.** The embodied AI community has provided various benchmarks in simulation platforms for tasks such as navigation [1, 81–84], object manipulation [85, 35, 86, 87], instruction following [6, 88–90], room rearrangement [28, 91], grasping [92] and SLAM [93]. While these benchmarks ensure reproducibility and fair comparison of different methods, there is always a gap between simulation and reality since it is infeasible to model all details of the real world in simulation. Our benchmark, in contrast, enables fair comparison of different methods and reproducibility of the results in the real world. Additionally, previous benchmarks often operate in a simplified discrete action space [20, 6], even forcing that structure on the real world [2].

**Robotics benchmarking.** Robotics benchmarks must contend with the diversity of hardware, morphology, and resources across labs. One solution is simulation [87, 59, 35, 20, 21, 83, 86, 6], which can provide reproducible and fair evaluations. However, the sim-to-real gap means simulation results may not be indicative of progress in the real world [2]. Another proposed solution is robotic competitions such as RoboCup@Home [3], the Amazon Picking Challenge [39], and the NIST task board [56]. However, participants typically use their own hardware, making it difficult to conduct fair comparisons of the different underlying methods, and means results are not transferable to different labs or settings. This is also a large barrier to entry to these competitions.

**Exploration of unseen environments.** Various papers have looked at object search in different home environments. Gervet et al. [2] use a mixture of heuristic/model-based planning and reinforcement learning to achieve strong results in a variety of real-world environments; importantly their planning-based methods perform competititvely to the best learned methods, and much better than end-to-end reinforcement learning on real tasks. Also promising is graph-based exploration. Kurenkov et al. [94] propose *hierarchical mechanical search*, based on a 3d scene graph representation, for exploring environments; although unlike in our Open-Vocabulary Mobile Manipulation task, they assume such a scene graph exists. Similarly, SayPlan [95] performs search over a compex scene graph using a large language model; however, this approach also does not look into iteratively constructing this scene graph on new scenes. While there is active work in iteratively exploring and building scene graphs and other hierarchical representations (e.g. [96]), there are not yet strongly established methods in this space.

# B  Limitations

Due to simulation limitations, we don't physically simulate grasping in the current version of the benchmark, which is why we provide a separate policy for this in the real world. Grasping is a well-studied problem [97–99], but simulations that train useful real-world grasp systems require special consideration. We also currently consider full natural language queries out-of-scope. Finally,

we did not evaluate many motion planners (see Sec. E.2), or performed task-and-motion-planning with replanning, as would be ideal for a long horizon task [100].

# C   Metrics

We informally defined our scoring metrics in Sec. 3. Here, we provide formal definitions of our partial success metrics.

## C.1   Simulation Success Metrics

Success in simulation is defined per stage as:

• **FindObj:** Successful if the agent reaches within 0.1m of a viewpoint of the target `object` on `start_receptacle`, and at least $0.1\%$ of the pixels in its camera frame belong to an `object` instance.

• **Pick:** Successful if **FindObj** succeeded, the agent enables the gripper at an instant where an `object` instance is visible and its end-effector reaches within 0.8m of a target object. We magically snap the `object` to the agent's gripper in simulation.

• **FindRec:** Successful if **Pick** succeeded, and the agent reaches within 0.1m of a viewpoint of a `goal_receptacle`, and at least $0.1\%$ of the pixels in its camera frame belong to the object containing a valid receptacle.

• **Place:** Successful if **FindRec** succeeded, the agent releases the `object` and subsequently the `object` stays in contact with the `goal_receptacle` with linear and angular velocities below a threshold of $5e{-}3$ m/s and $5e{-}2$ rad/s respectively for 50 contiguous steps. Further, the agent should not collide with the scene while attempting to place the object.

An episode is considered to have succeeded if it succeeds in all 4 stages within 1250 steps.

**Better pick success condition.** We plan to use a more realistic grasping condition in simulation. We try replacing the magic snap in simulation with a stricter condition that requires the agent to move its arm near the object without colliding with the scene or other objects. Additionally, we tested a baseline (Figure 5) that performs top-down grasps resembling our real-world grasping policy, and resorting to side-ways grasps when the object is farther. While this baseline succeeds in reaching the object starting from an object viewpoint 79% of the time, it does so without colliding only 47% of the time.

## C.2   Real World Success Metrics

Success in real world is defined per stage as:

• **FindObj:** Successful if the agent reaches within 1m of the target `object` on `start_receptacle` and the `object` is visible in the RGB image from the camera.

• **Pick:** Successful if **FindObj** succeeded and the agent successfully picks up the `object` from the `start_receptacle`.

• **FindRec:** Successful if **Pick** succeeded, and the agent reaches within 1m of a `goal_receptacle`, and the `goal_receptacle` is visible in the RGB image from the camera.

• **Place:** Successful if **FindRec** succeeded and the agent places `object` on a `goal_receptacle` and the object settles down on the `goal_receptacle` stably.

Given that the scene we use in the real world is much smaller than the apartments in the simulation, we allow the agent to act in the environment for 300 timesteps. The episode is considered to have succeeded if it succeeds in all 4 stages.

Top-down grasp success | Sideways grasp success | Grasp failures

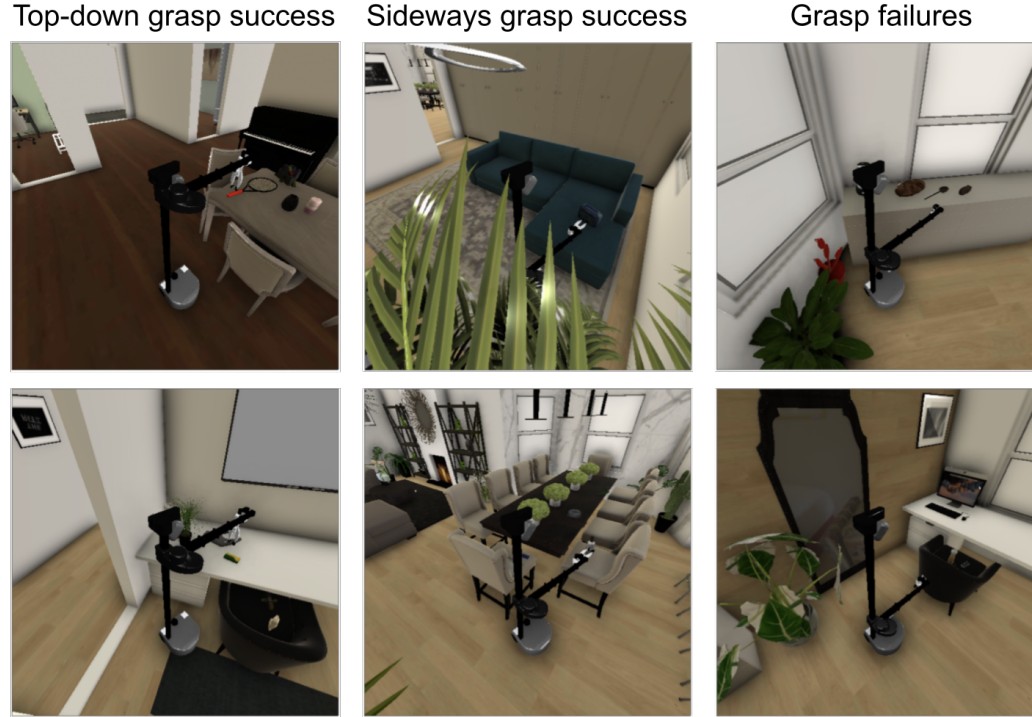

Figure 5: A few success and failure cases for our simple grasping policy under the new grasp success condition that requires the agent's arm to reach near the object without colliding. The agent resorts to sideways grasps when the object can't be reached via top-down grasp that bends the gripper. Most grasping failures are because of the collisions with the scene.

# D   Simulation Details

## D.1   Object Categories Appearing in the Scene Dataset

```
action_figure, android_figure, apple, backpack, baseballbat, basket, basketball,
bath_towel, battery_charger, board_game, book, bottle, bowl, box, bread, bundt_pan,
butter_dish, c-clamp, cake_pan, can, can_opener, candle, candle_holder, candy_bar,
canister, carrying_case, casserole, cellphone, clock, cloth, credit_card, cup,
cushion, dish, doll, dumbbell, egg, electric_kettle, electronic_cable, file_sorter,
folder, fork, gaming_console, glass, hammer, hand_towel, handbag, hard_drive, hat,
helmet, jar, jug, kettle, keychain, knife, ladle, lamp, laptop, laptop_cover,
laptop_stand, lettuce, lunch_box, milk_frother_cup, monitor_stand, mouse_pad,
multiport_hub, newspaper, pan, pen, pencil_case, phone_stand, picture_frame,
pitcher, plant_container, plant_saucer, plate, plunger, pot, potato, ramekin,
remote, salt_and_pepper_shaker, scissors, screwdriver, shoe, soap, soap_dish,
soap_dispenser, spatula, spectacles, spicemill, sponge, spoon, spray_bottle,
squeezer, statue, stuffed_toy, sushi_mat, tape, teapot, tennis_racquet,
tissue_box, toiletry, tomato, toy_airplane, toy_animal, toy_bee, toy_cactus,
toy_construction_set, toy_fire_truck, toy_food, toy_fruits, toy_lamp, toy_pineapple,
toy_rattle, toy_refrigerator, toy_sink, toy_sofa, toy_swing, toy_table, toy_vehicle,
tray, utensil_holder_cup, vase, video_game_cartridge, watch, watering_can,
wine_bottle
```
In Fig. 7 we show some of the examples of a selection of these categories from the training and validation/test splits.

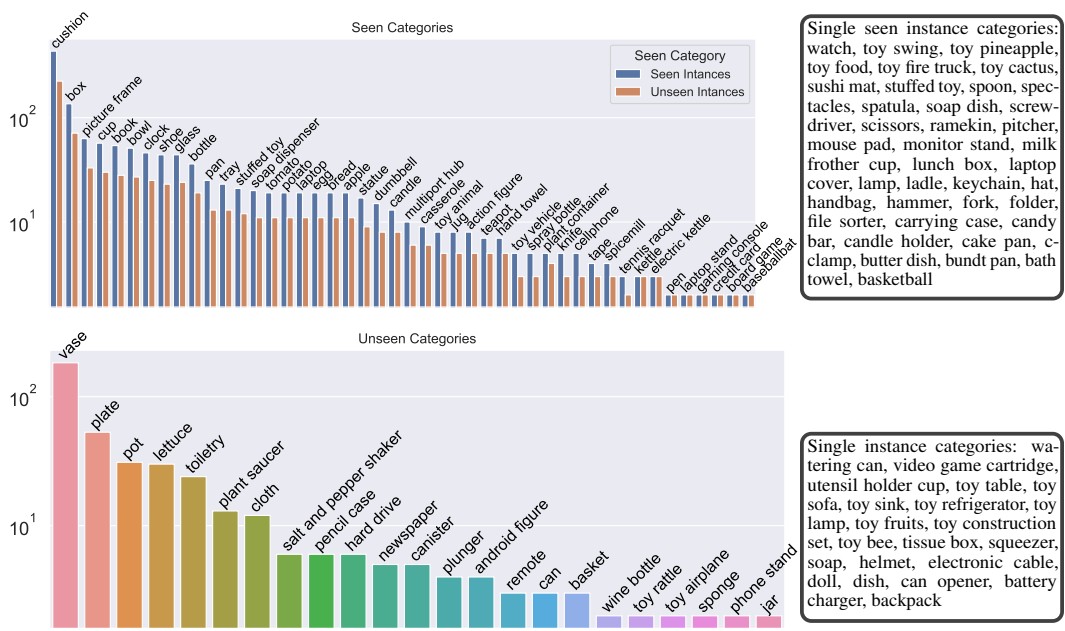

Figure 6: Number of objects across different splits, for both seen categories and unseen categories. We divide objects between categories which appear in training data – *seen categories* – and those that do not – *unseen categories*. The goal of Open-Vocabulary Mobile Manipulation is to be able to find and manipulate objects specified by language.

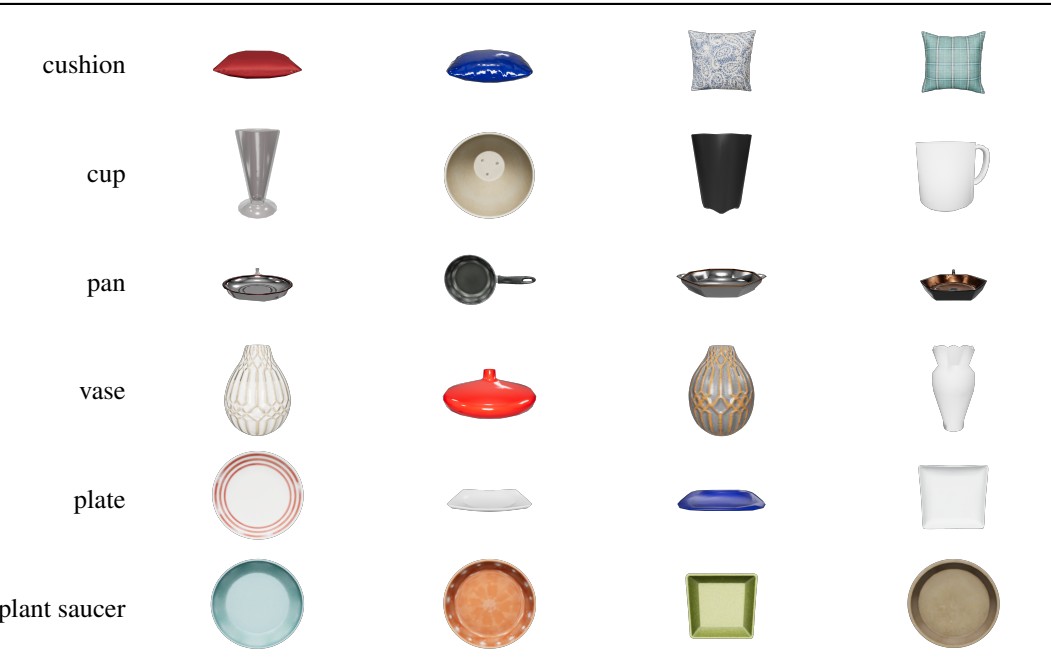

Figure 7: Example objects in our object dataset across 6 categories. The cushion, cup, and pan categories are in the train split, and the vase, plate, and plant saucer are in the validation and test sets.

## D.2 Episode Generation Details

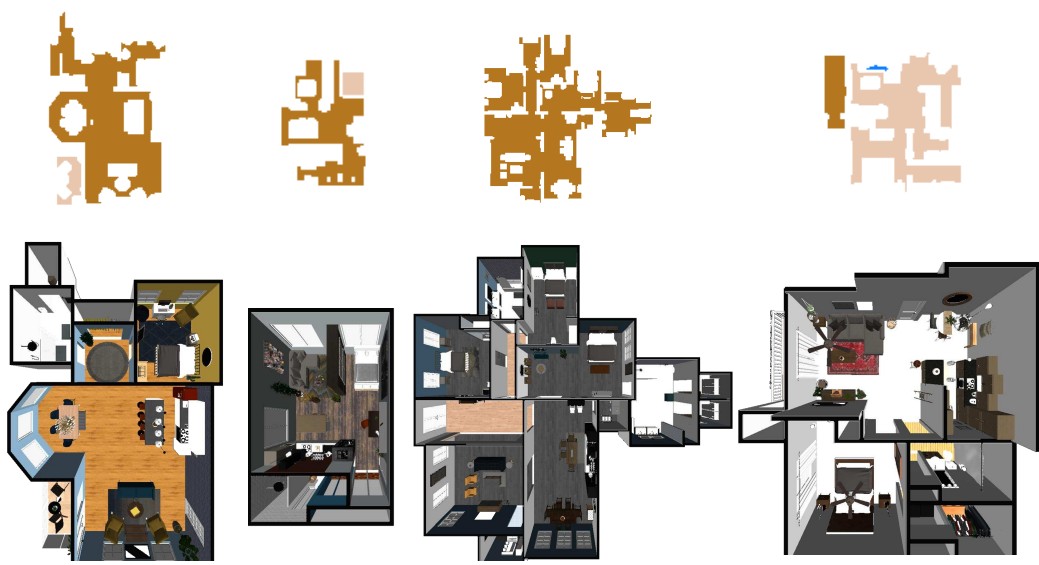

Figure 8: Visualization of the navigable geometry (top row) and top-down views of example scenes from the Habitat Synthetic Scenes Dataset (HSSD) [19]. We use the computed navigable area to efficiently generate a large number of episodes for the Open-Vocabulary Mobile Manipulation task. Object placement positions are sampled to be near navigable areas of the map, atop one of a large variety of different receptacles, such that the robot can reach them.

When generating episodes, we find the largest indoor navigable area in each scene, and then filter the list of all receptacles from this scene that are too small for object placement. Fig. 8 shows the navigable islands in several of our scenes (top row), and corresponding top-down views of each scene in the bottom row. We then sample objects according to the current split (train, validation, or test). We run physics to ensure that objects are placed in stable locations. Then we select objects randomly from the appropriate set, as determined by the current split.

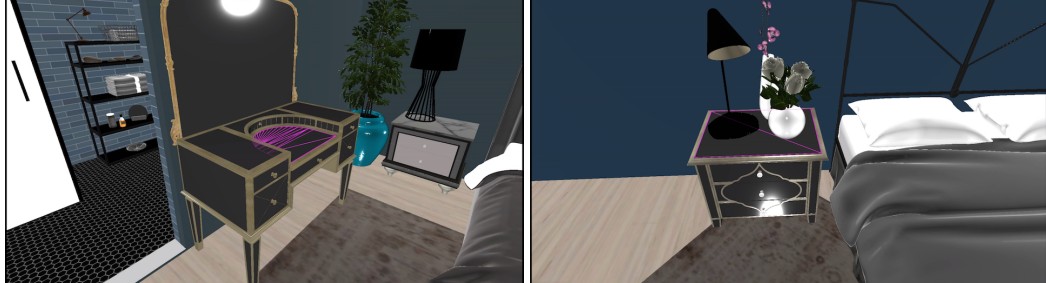

Figure 9: First-person view from different precomputed viewpoints in our episode dataset. These viewpoints are used as goals for training navigation skills, and are used in the initialization of the placement and gaze/grasping skills as well. The purple mesh indicates receptacle surface.

Finally, we generate a set of candidate viewpoints, shown in Fig. 9, which represent navigable locations to which the robot can move for each receptacle. These are used for training specific skills, such as navigation to receptacles. Each viewpoint corresponds to a particular `start_receptacle` or `goal_receptacle`, and represents a nearby location where the robot can see the receptacle and is within 1.5 meters. Fig. 10 gives examples of where these viewpoints are created.

**Navmesh:** We precompute a navigable scene geometry as done in [20] for faster collision checks of the agent with the scene. The "mesh" comprising this navigable geometry is referred to as a navmesh.

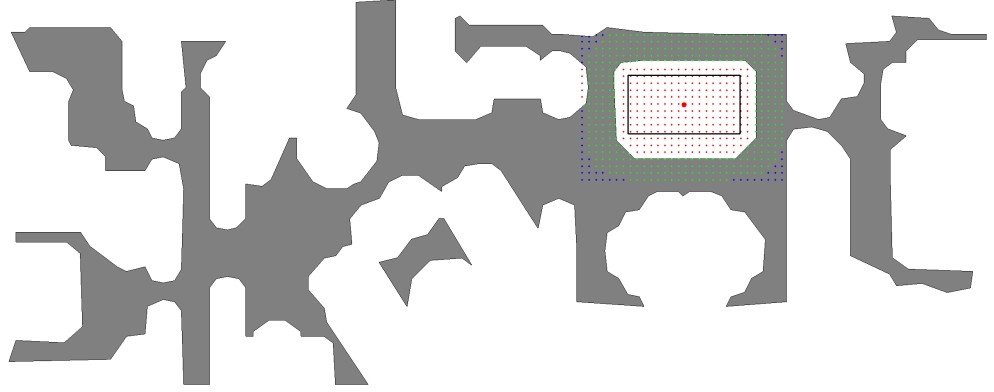

Figure 10: Viewpoints created for an object during episode generation. The gray area is the navigable region of the scene. The big red dot and the black box are the object's center and bounding box respectively. The surrounding dots are viewpoint candidates: red dots were rejected because they weren't navigable, and blue dots were rejected because they were too far from the object. The green dots are the final set of viewpoints.

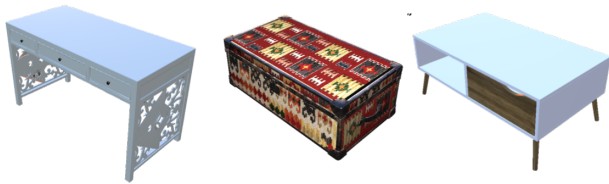

Figure 11: The variation in instances belonging to the "table" category in our dataset.

**Number of objects:** This is dynamically set per scene to 1.5-2× the total available receptacle area in $m^2$. For example, if the total receptacle surface area for a scene is $10m^2$, then 15-20 objects will be placed. The exact number of objects will be randomly selected per episode to be in this range.

The full set of included receptacles in simulation is: `bathtub, bed, bench, cabinet, chair, chest_of_drawers, couch, counter, filing_cabinet, hamper, serving cart, shelves, shoe_rack, sink, stand, stool, table, toilet, trunk, wardrobe,` & `washer_dryer`.

### D.3   Diversity in Receptacle Instances

The instances within each receptacle category exhibit substantial variability. Figure 11 shows a few different receptacles from our dataset belonging to the "table" category.

### D.4   Scene Clutter Complexity

Our procedural placement of target and distractor objects creates diverse and interesting scenarios that require reasoning over which direction to approach receptacles, stable placement in clutter, open vocabulary object detection under occlusion which makes the task quite challenging. Figure 12 shows a few examples of clutter surrounding target objects in our scenes.

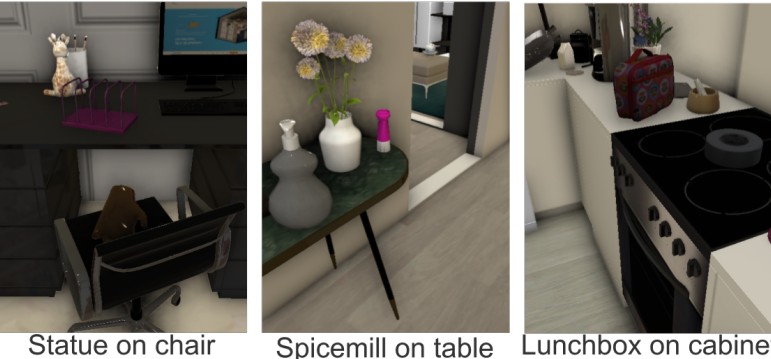

Statue on chair    Spicemill on table    Lunchbox on cabinet

Figure 12: A few examples of clutter surrounding the target object in our simulation settings.

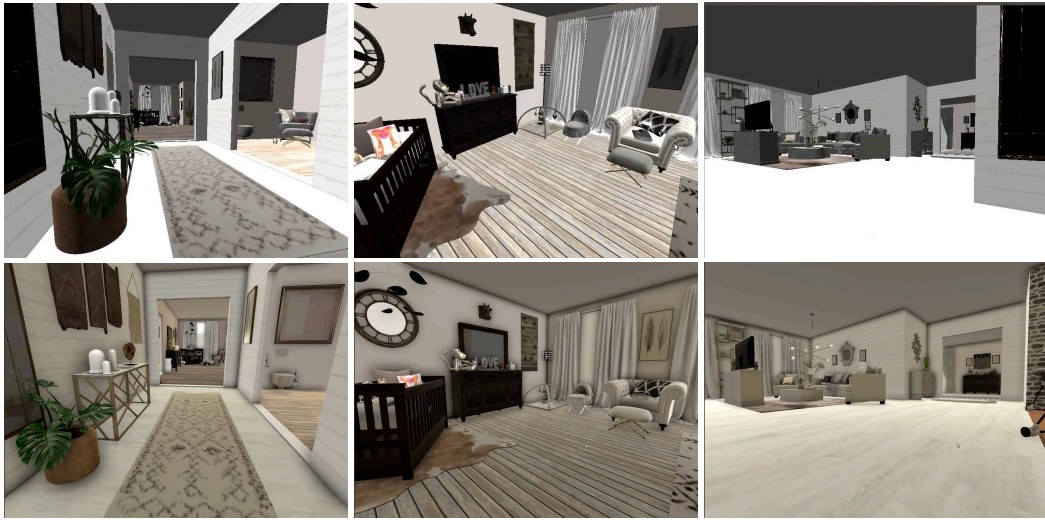

Figure 13: Here we present the improvements in scene visuals with Horizon-based Ambient Occlusion (HBAO) and expanded Physics-based Rendering (PBR) material support added to the Habitat renderer. The top row shows images from the default renderer whereas the bottom row shows the improved renderings.

## D.5 Improved scene visuals

We rewrote and expanded the existing Physically-Based Rendering shader (PBR) and added Horizon-based Ambient Occlusion (HBAO) to the Habitat renderer, which led to notable improvements in viewing quality which were necessary for using the HSSD [19] dataset.

- Rewrote PBR and Image Based Lighting (IBL) base calculations.
- Added multi-layer material support covering `KHR_materials_clearcoat`, `KHR_materials_specular`, `KHR_materials_ior`, and `KHR_materials_anisotropy` for both direct and indirect (IBL) lighting.
- Added tangent frame synthesis if precomputed tangents are not provided.
- Added HDR Environment map support for IBL.

We present comparisons between default Habitat visuals and improved renderings in Figure 13.

We also benchmark the ObjectNav training speeds of a DDPPO-based RL agent with and without the improved rendering and present the results in 14. We see that the improvement in scene lighting and

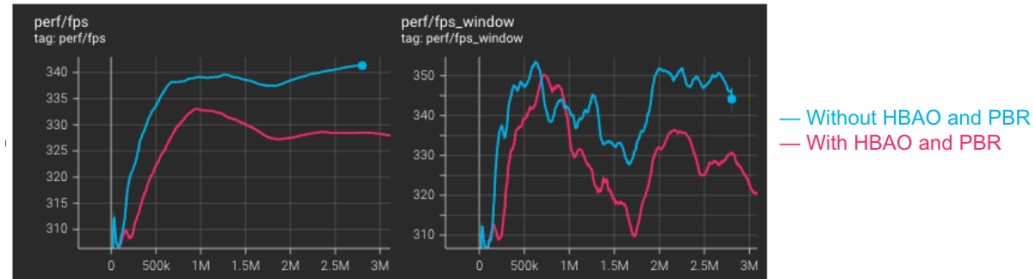

Figure 14: **Minor drop in FPS with improved scene rendering**: Here, we benchmark the training speeds (through FPS numbers) of two ObjectNav training runs with and without the HBAO and PBR-based improved scene visuals. We observe that the improved rendering leads to a very small drop in FPS from around 340 to 330 (3 % drop).

rendering comes at the cost of only a 3% dip in training FPS (decreasing from around 340 to around 330).

### D.6  Action Space Implementation

We look at two different choices of action space for our navigation agents, either making discrete or continuous predictions about where to move next. Our expectation from prior work might be that the discrete action space would be notably easier for agents to work with.

**Discrete.** Previous benchmarks often operate in a fully discrete action space [20, 6], even in the real world [2]. We implement a set of discrete actions, with fixed in-place rotation left and right, and translation of steps $0.25m$ forward.

**Continuous.** Our continuous action space is implemented as a teleporting agent, where the robot needs to move around by predicting a local waypoint. Our robot's low-level controllers are expected to be able to get the robot to this location, in lieu of simulating full physics for the agent.

In simulation, this is implemented as a check against the navmesh - we use the navmesh to determine if the robot will go into collision with any objects if moved towards the new location, and move it to the closest valid location instead.

## E  HomeRobot Implementation Details

Here, we describe more specifics for how we implemented the heuristic policies provided as a baseline to accelerate home assistant robot research.

Although there exists a considerable body of prior research looking at learning specific grasping [101, 98, 99, 97] or placement [102, 17] skills, we found that it was easiest to implement heuristic policies with low CPU/GPU requirements and high interpretability. Other recent works have similarly used heuristic grasping and placement policies to great affect (e.g. TidyBot [62]).

There are three different repositories within the open-source HomeRobot library:

- `home_robot`: Shared components such as Environment interfaces, controllers, detection and segmentation modules.

- `home_robot_sim`: Simulation stack with Environments based on Habitat.

- `home_robot_hw`: Hardware stack with server processes that runs on the robot, client API that runs on the GPU workstation, and Environments built using the client API.

Most policies are implemented in the core `home_robot` library. Within HomeRobot, we also divide functionality between **Agents** and **Environments**, similar to how many reinforcement learning benchmarks are set up [20].

- **Agents** contain all of the necessary code to execute policies. We implement agents which use a mixture of heuristic policies and policies learned on our scene dataset via reinforcement learning.

- **Environments** provide common logic; they provide **Observations** to the Agent, and a function which allows them to apply their action to the (real or simulated) environment.

## E.1 Pose Information

We get the global robot pose from Hector SLAM [103] on the Hello Robot Stretch [22], which is used when creating 2d semantic maps for our model-based navigation policies.

## E.2 Low-Level Control for Navigation

The Hello Stretch software provides a native interface for controlling the linear and angular velocities of the differential-drive robot base. While we do expose an interface for users to control these velocities directly, it is desireable to have desired short-term goals as a more intuitive action space for policies, and to make them update-able at any instant to allow for replanning.

Thus, we implemented a velocity controller that produces continuous velocity commands that moves the robot to an input goal pose. The controller operates in a heuristic manner: by rotating the robot so that it faces the goal position at all times while moving towards the goal position, and then rotating to reach the goal orientation once goal position is reached. The velocities to induce these motions are inferred with a trapezoidal velocity profile and some conditional checks to prevent it from overshooting the goal.

**Limitations** The Fast Marching Method-based motion planning from prior work [2] that we describe in Sec. E.2. It assumes the agent is a cylinder, and therefore is much more limited in where it can navigate than, e.g., a sampling based motion planner like RRT-connect [104] which can take orientation into account. In addition, our semantic mapping requires a list of classes for use with DETIC [27]; instead, it would be good to use a fully open-vocabulary scene representation like CLIP-Fields [11], ConceptFusion [15], or USA-Net [12], which would also improve our motion planning significantly.

## E.3 Heuristic Grasping Policy

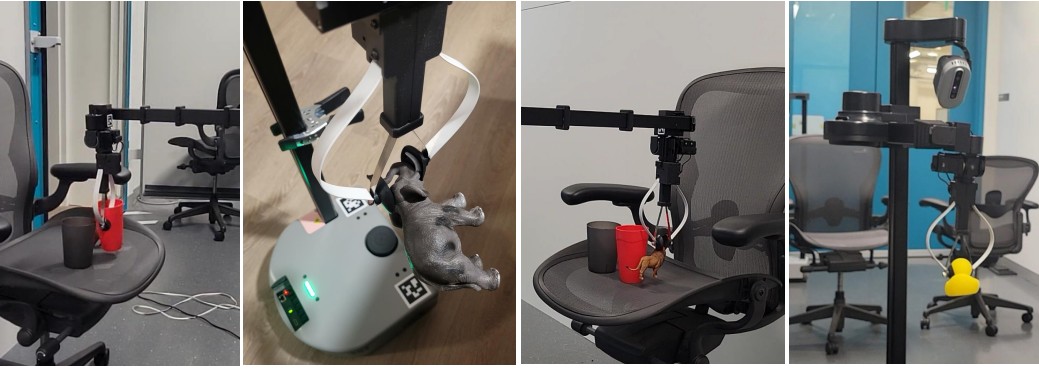

Figure 15: Grasping tests in various lab environments. To provide a strong baseline, we tuned the grasp policy to be highly reliable given the Stretch's viewpoint, on a variety of objects.

Numerous powerful grasp generation models have been proposed in the literature, such as GraspNet-1Billion [99], 6-DOF GraspNet [98], and Contact-GraspNet [97]. However, for transparency, repro-

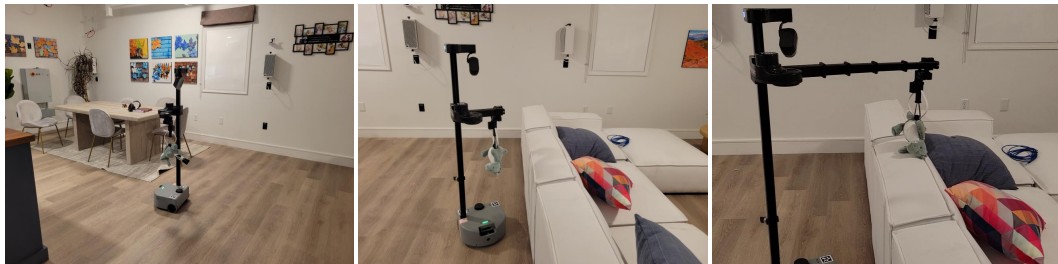

Figure 16: An example of the robot navigating to a `goal_receptacle` (sofa) and using the heuristic place policy to put down the `object` (stuffed animal). Heuristic policies provide an interpretable and easily extended baseline.

ducibility, and ease of installation, we implement a simple, heuristic grasping policy, which assumes a parallel gripper performing top-down grasps. Heuristic grasp policies appear throughout robotics research (e.g. in TidyBot [62]). In our case, the heuristic policy voxelizes the point cloud, and chooses areas at the top of the object where points exist, surrounded by free space, in order to grasp. Fig. 15 shows the simple grasp policy in action and additional details are presented in Sec. E.3. This policy works well on a wide variety of objects, and we saw comparable performance to the state-of-the-art open-source grasping models we tested [97, 99].

The intuition is to identify areas where the gripper fingers can descend unobstructed into two sides of a physical part of the object, which we do through a simple voxelization scheme. We take the top 10% of points in an object, voxelize at a fixed resolution of 0.5cm, and choose grasps with free voxels (where fingers can go) on either side of occupied voxels. In practice, this achieved a high success rates on a variety of real objects.

The procedure is as follows:

1. Given a target object point cloud, convert the point cloud into voxels of size 0.5 cm.
2. Select top 10% occupied voxels with the highest Z coordinates.
3. Project the selected voxels into a 2-D grid.
4. Consider grasps centered around each occupied voxel, and identify three regions: two where the gripper fingers will be and one representing the space between the fingers.
5. Score each grasp based on 1) how occupied the region between the fingers is, and 2) how empty the two surrounding regions are.
6. Perform smoothing on the grasp scores to reject outliers (done by multiplying scores with adjacent scores).
7. Output grasps with final scores above some threshold.

We compared this policy to other methods like ContactGraspnet [97], 6-DoF Graspnet [98, 101], and Graspnet 1-Billion [99]. We saw more intermittent failures due to sensor noise using these pretrained methods, even after adapting the grasp offsets to fit to the Hello Robot Stretch's gripper geometry. In the end, we leave training better grasp policies to future work.

### E.4 Heuristic Placement Policy

As with grasping, a number of works on stable placement of objects have been proposed in the literature [102, 17]. To provide a reasonable baseline, we implement a heuristic placement strategy that assumes that the end-receptacle is at least barely visible when it takes over; projects the segmentation mask onto the point cloud and chooses a voxel on the top of the object. Fig. 16 shows an example of the place policy being executed in the real world.

Specifically, our heuristic policy is implemented as such:

1. Detect the end-receptacle in egocentric RGB observations (using DETIC [27]), project predicted image segment to a 3D point cloud using depth, and transform point cloud to robot base coordinates using camera height and tilt.

2. Estimate placement point: Randomly sample 50 points on the point cloud and choose one that is at the center of the biggest (point cloud) slab for placing objects. This is done by scoring each point based on the number of surrounding points in the X/Y plane (Z is up) within a 3 cm height threshold.

3. Rotate robot for it to be facing the placement point, then move robot forward if it is more than 38.5 cm away (length of retracted arm + approximate length of the Stretch gripper).

4. Re-estimate placement point from this new robot position.

5. Accordingly, set arm's extension and lift values to have the gripper be a few cm above placement position. Then, release the object to land on the receptacle.

### E.5 Navigation Planning

Our heuristic baseline extends prior work [2], which was shown to work in a wide range of human environments. We tune it for navigating close to other objects and extended it to work in our continuous action space – challenging navigation aspects not present in the original paper. The baseline has three components:

**Semantic Mapping Module.** The semantic map stores relevant objects, explored regions, and obstacles. To construct the map, we predict semantic categories and segmentation masks of objects from first-person observations. We use Detic [27] for object detection and instance segmentation and backproject first-person semantic segmentation into a point cloud using the perceived depth, bin it into a 3D semantic voxel map, and finally sum over the height to compute a 2D semantic map.

We keep track of objects detected, obstacles, and explored areas in an explicit metric map of the environment from [105]. Concretely, it is a binary $K$ x $M$ x $M$ matrix where $M$ x $M$ is the map size and $K$ is the number of map channels. Each cell of this spatial map corresponds to 25 cm$^2$ (5 cm x 5 cm) in the physical world. Map channels $K = C + 4$ where $C$ is the number of semantic object categories, and the remaining 4 channels represent the obstacles, the explored area, and the agent's current and past locations. An entry in the map is one if the cell contains an object of a particular semantic category, an obstacle, or is explored, and zero otherwise. The map is initialized with all zeros at the beginning of an episode and the agent starts at the center of the map facing east.

**Frontier Exploration Policy.** We explore the environment with a heuristic frontier-based exploration policy [106]. This heuristic selects as the goal the point closest to the robot in geodesic distance within the boundary between the explored and unexplored region of the map.

**Navigation Planner.** Given a long-term goal output by the frontier exploration policy, we use the Fast Marching Method [107] as in [105] to plan a path and the first low-level action along this path deterministically. Although the semantic exploration policy acts at a coarse time scale, the planner acts at a fine time scale: every step we update the map and replan the path to the long-term goal. The robot attempts to plan to goals if they have been seen; if it cannot get within a certain distance of the goal objects, then it will instead plan to a point on the frontier.

**Navigating to objects on `start_receptacle`.** Since small objects (*e.g.* `action_figure`, `apple`) can be hard to locate from a distance, we leverage the typically larger `start_receptacle` goals for finding objects. We make the following changes to the original planning policy [108]:

1. If object and `start_receptacle` co-occur in at least one cell of the semantic map, plan to reach the object

2. If the object is not found but `start_receptacle` appears in the semantic map after excluding the regions within 1m of the agent's past locations, plan to reach the `start_receptacle`

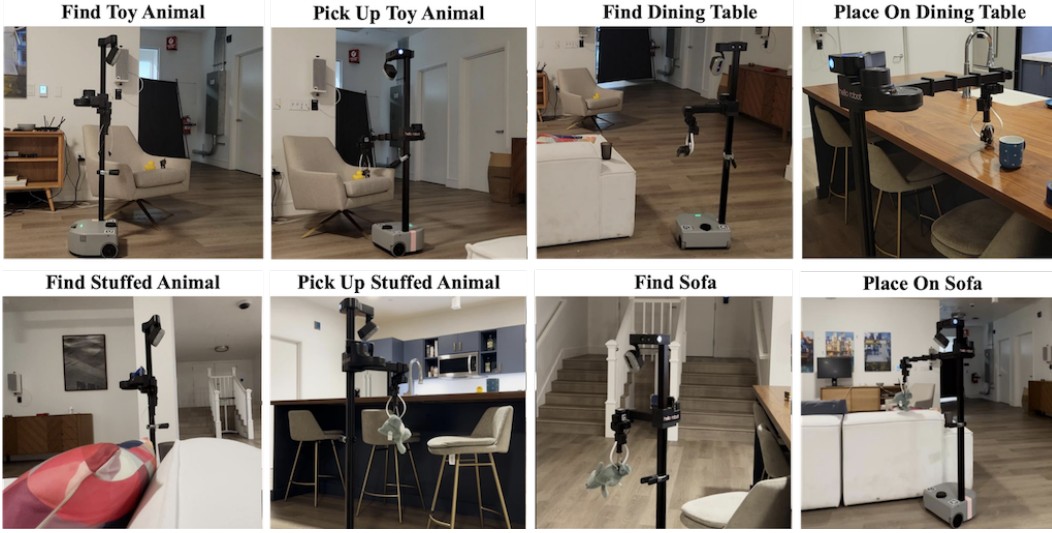

Figure 17: Real-world examples (also see Fig 2). Our system is able to find held-out objects in an unseen environment and navigate to receptacles in order to place them, all with no information about the world at all, other than the relevant classes. However, we see this performance is highly dependent on perception performance for now; many real-world examples also fail due to near-miss collisions.

3. Otherwise, plan to reach the closest frontier

In step 2, we exclude the regions that the agent has been close to, to prevent it from re-visiting already visited instances of `start_receptacle`.

### E.6  Navigation Limitations

We implemented a navigation system that was previously used in extensive real-world experiments [2], but needed to tune it extensively for it to get close enough to objects to grasp and manipulate them. The original version by Gervet et al. [2] was focused on finding very large objects from a limited set of only six classes. Ours supports many more, but as a result, tuning it to both be able to grasp objects and avoid collisions in all cases is difficult.

This is partly because the planner is a discrete planner based on the Fast Marching Method [107], which cannot take orientation into account and relies on a 5cm discretization of the world. ampling-based motion planners like RRT-Connect [104], or like that used in the Task and Motion Planning literature [100, 8], may offer better solutions. Alternately, we could explore optimization-based planners specifically designed for open-vocabulary navigation planning, as has recently been proposed [12].

Our navigation policy relies on accurate pose information from Hector SLAM [103], and unfortunately does not handle dynamic obstacles. It also models the robot's location as a cylinder; the Stretch's center of rotation is slightly offset from the center of this cylinder, which is not currently accounted for. Again, sampling-based planners might be better here.

## F  Reinforcement Learning Baseline

We train four different RL policies: `FindObject`, `FindReceptacle`, `GazeAtObject`, and `PlaceObject`.

### F.1 Action Space

#### F.1.1 Navigation Skills

`FindObject` and `FindReceptacle` are, collectively, navigation skills. For these two skills, we use the discrete action space, as mentioned in Sec. D.6. In our experiments, we found the discrete action space was better at exploration and easier to train.

#### F.1.2 Manipulation Skills

For our manipulation skills, we using a continuous action space to give the skills fine grained control. In the real world, low-level controllers have limits on the distance the robot can move in any particular step. Thus, in simulation, we limit our base action space by only allowing forward motions between 10-25 cm, or turning by 5-30 degrees in a single step. The head tilt, pan and gripper's yaw, roll and pitch can be changed by at most 0.02-0.1 radians in a single step. The arm's extension and lift can be changed by at most 2-10cm in a single step. We learn by *teleporting* the base and arm to the target locations.

### F.2 Observation Space

Policies have access to depth from the robot head camera, and semantic segmentation, as well as the robot's pose relative to the starting pose (from SLAM in the real world), camera pose, and the robot's joint states, including the gripper. RGB image is available to the agent but not used during training.

### F.3 Training Setup

All skills are trained using a slack reward of -0.005 per step, incentivizing completion of task using minimum number of steps. For faster training, we learn our policies using images with a reduced resolution of 160x120 (compared to Stretch's original resolution of 640x480).

#### F.3.1 Navigation Skills

We train `FindObject` and `FindReceptacle` policies for the agent to reach a candidate object or a candidate target receptacle respectively. The training procedure is the same for both skills. We pass in the CLIP [14] embedding corresponding with the goal object, as well as segmentation masks corresponding with the detected target objects. The agent is spawned arbitrarily, but at least 3 meters from the target, and must move until within 0.1 meters of a goal "viewpoint," where the object is visible.

**Input observations:** Robot head camera depth, ground-truth semantic segmentation for all receptacle categories (receptacle segmentation), robot's pose relative to the starting pose, joint sensor giving states of camera and arm joints. We implement object-level dropout for the semantic segmentation mask, where each object has a probability of 0.5 of being left out of the mask. In addition, the input observation space includes the following:

- **Goal specification:** For `FindObject`, we pass in the CLIP embedding of the target object and the start receptacle category. For `FindReceptacle`, we pass in the goal receptacle category.
- **Goal segmentation images:** During training, the simulator provides ground truth goal object segmentation; on the real robot, these are predicted by DETIC [27]. For `FindObject`, we pass in two channels: one showing all instances of candidate objects, one showing all instances of candidate start receptacles. For `FindReceptacle`, we pass a single channel showing all instances of candidate goal receptacles. We implement a similar object-level dropout procedure here as we did for the receptacle segmentation.

**Initial state:** The agent is spawned at least 3m away from candidate object or receptacle. It starts in "navigation mode," with the robot's head facing forward.

**Actions:** The policy predicts translation and rotation waypoints, as well as a discrete stop action.

**Success condition:** The agent should call the discrete stop action when it reaches within 0.5m of a goal view point. The agent should be facing the target: the angle between agent's heading direction and the ray from robot to center of the closest candidate object should be no more than 15 degrees.

**Reward:** Assume at time step $t$, the geodesic distance to the closest goal is given by $d(t)$, the angle between agent's heading direction and the ray from agent to closest goal is given by $\theta(t)$, and did_collide($t$) indicates if the action the agent took at time $t - 1$ resulted in a collision at time $t$. The training reward is given by:

$$R_{FindX}(t) = \alpha[d(t-1) - d(t)] + \beta\mathbb{1}[d(t) \leq D_{close}][\theta(t-1) - \theta(t)] + \gamma\mathbb{1}[\text{did\_collide}(t)]$$

with $\alpha = 1$, $\beta = 1$, $\gamma = 0.3$ and $D_{close} = 3$.

### F.3.2 `GazeAtObject`

The `GazeAtObject` skill starts near the object, and provides some final refinement steps until the agent is close enough to call a grasp action, i.e. it is in arm's length of the object and the object is centered and visible. The agent needs to move closer to the object and then adjust its head tilt until the candidate object is close and centered. It makes predictions to move and rotate the head, as well as to center the object and make sure it's within arm's length, so that the discrete grasping policy can execute.

The `GazeAtObject` skill is supposed to start off from locations and help reach a location within arm's length of a candidate object. This is trained by first initialising the agents close to candidate start receptacles. The agent is then tasked to reach close to the agent and adjust its head tilt such that the candidate object is close and centered in the agent's camera view. We next provide details on the training setup.

**Input observations:** Ground truth semantic segmentation of candidates objects, head depth sensor, joint sensor giving all head and arm joint states, sensor indicating if the agent is holding any object, clip embedding for the target object name.

**Initial state:** The robot again starts in "navigation mode," with its arm retracted, with the gripper facing downwards, and with the head/camera facing the base, base at an angle of 5 degrees of the center object and on one of the "viewpoint" locations pre-computed during episode generation. The object will therefore be assumed to be visible.

**Actions:** This policy predicts base translation and rotation waypoints, camera tilt, as well as a discrete "grasp" action.

**Success condition:** The center pixel on the camera should correspond to a valid candidate object and the agent's base should be within 0.8m from the object.

**Reward:** We train the gaze-policy mainly with a dense reward based on distance to goal. Specifically, assuming the distance of the end-effector to the closest candidate goal at time $t$ is $d(t)$ (in metres), the agent receives a reward proportional to $d(t-1) - d(t)$. Further, when the agent reaches with 0.8m, we provide an additional reward for incentivizing the agent to look towards the object.

Let $\theta(t)$ denote the angle (in radians) between the ray from agent's camera to the object and camera's normal. Then the reward is given as:

$$R_{Gaze}(t) = \alpha[d(t-1) - d(t)] + \beta\mathbb{1}[d(t) \leq \gamma]cos(\theta(t))$$

with $\alpha = 2$, $\beta = 1$ and $\gamma = 0.8$ in our case.

The agent receives an additional positive reward of 2 once the episode succeeds and receives a negative reward of $-0.5$ for centering its camera towards a wrong object.

### F.3.3 `PlaceObject`

Finally, the robot must move its arm in order to place the object on a free spot in the world. In this case, it starts at a viewpoint near a `goal_receptacle`. It must move up to the object and open its gripper in order to place the object on this surface.

**Input observations:** Ground truth segmentation of goal receptacles, head depth sensor, joint sensor, sensor indicating if the agent is holding any object, CLIP [14] embedding for the name of object being held.

**Initial configuration:** Arm retracted, with gripper down and holding onto an object, head facing the base. The agent is spawned on a viewpoint with its base facing the object with an error of at most 15 degrees.

**Actions:** Base translation and rotation waypoints, all arm joints (arm extension, arm lift, gripper yaw, pitch and roll), a manipulation mode action that can be invoked only once in an episode to turn the agent's head towards the arm and rotate the base left by 90 degrees. The agent is not allowed to move its base while in manipulation mode.

**Success condition:** The episode succeeds if the agent releases the object and the object stays on the receptacle for 50 timesteps.

**Reward:** The agent receives a positive sparse reward of 5 when it releases the object and the object comes in contact with a target receptacle. Additionaly, we provide a positive reward of 1 for each step the object stays in contact with the target receptacle. It receives a negative reward of $-1$ if the agent releases the object but the object does not come in contact with the receptacle.

### F.4 ConceptFusion

In the main paper, we introduced two key approaches based on end-to-end reinforcement learning and a heuristic baseline. Both methods are dependent on the detection results generated by a readily available open vocabulary object detector [27]. Notably, these 2D detection models do not leverage information from prior time steps to inform their detection decisions.

In order to address these limitations, we explored the application of ConceptFusion [15], an open-set scene representation technique. ConceptFusion harnesses foundation models like CLIP [14], DINO [109], and others to construct 3D maps from multiple images. For our experimentation, we employed the open-source implementation of ConceptFusion, which utilizes the Segment Anything Model (SAM) [110] for object segmentation in RGB images and CLIP for feature extraction from each segmentation mask. It's important to note that our experiments were conducted in a simulated environment, obviating the need for GradSLAM [111], as we had access to ground truth depth maps and pose information to support our map construction efforts.

During our initial experimentation, we observed that ConceptFusion demanded significant computational resources and memory, with processing times reaching up to 5 seconds per frame for map construction. Remarkably, it's worth noting that the authors of ConceptFusion have recently published a new paper titled "ConceptGraphs: Open-Vocabulary 3D Scene Graphs for Perception and Planning," [112] which addresses some of the computational challenges we encountered. However, we leave the exploration of ConceptGraphs as a potential avenue for future research.

## G  Additional Analysis

Here, we provide some additional analysis of the different skills we trained to complete the Open-Vocabulary Mobile Manipulation task.

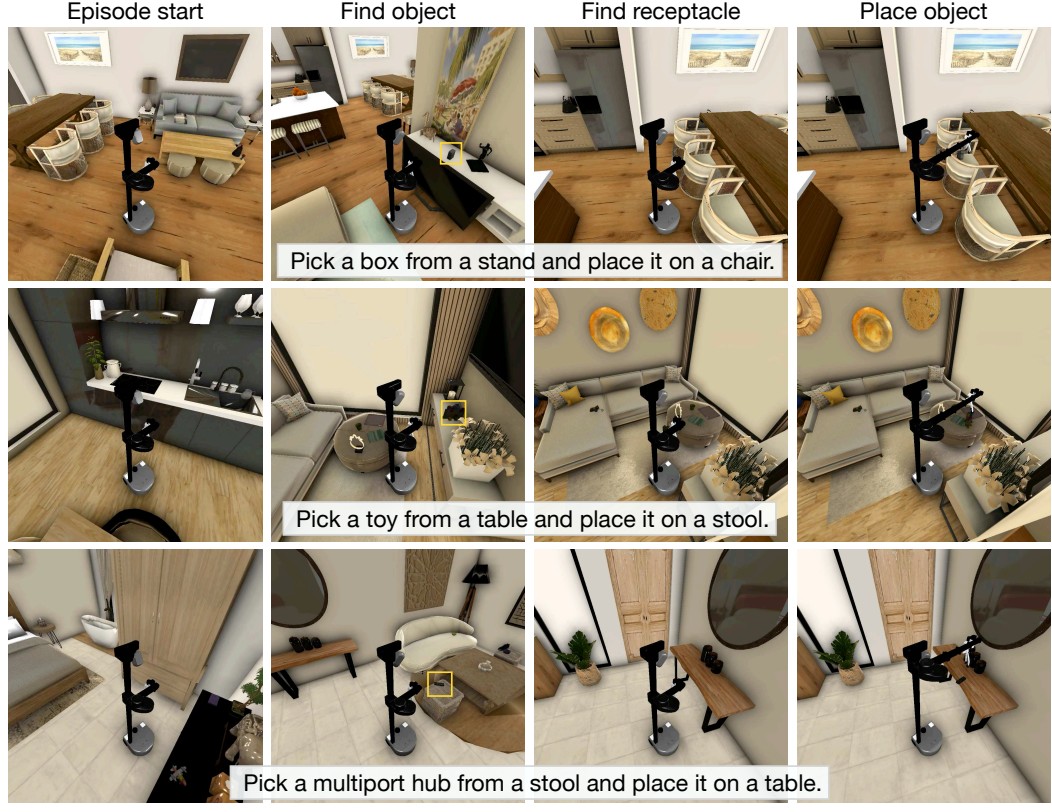

Figure 18: We show multiple executions of the Open-Vocabulary Mobile Manipulation task in a variety of simulated environments.

| Nav. | Manip. | Perception | FindObj | Gaze | FindRec | Place | Total |
|---|---|---|---|---|---|---|---|
| Heuristic | Heuristic | Ground Truth | 291.8 | - | 65.5 | 8.4 | 360.5 |
| Heuristic | RL | Ground Truth | 293.5 | 19.4 | 64.3 | 84.4 | 438.7 |
| RL | Heuristic | Ground Truth | 295.1 | - | 105.0 | 7.0 | 401.6 |
| RL | RL | Ground Truth | 302.4 | 25.7 | 112.8 | 45.9 | 455.2 |
| Heuristic | Heuristic | DETIC [27] | 335.0 | - | 29.5 | 6.7 | 361.8 |
| Heuristic | RL | DETIC [27] | 330.1 | 152.0 | 27.5 | 68.2 | 556.5 |
| RL | Heuristic | DETIC [27] | 509.5 | - | 153.3 | 7.1 | 610.4 |
| RL | RL | DETIC [27] | 539.1 | 101.3 | 124.4 | 33.7 | 634.7 |

Table 5: The number of steps that the agent takes performing each of the skills for different baselines. Note that here we only consider the cases where the skill terminates. The last column gives the total number of steps the agent takes on average for executing the four skills.

## G.1 Number of steps taken in each stage by different baselines

Table 5 shows the number of steps taken by each skill in our baseline. With DETIC perception, we observed that the RL skills explored less efficiently than our simple heuristic-based planner; this translates to far fewer steps taken in successful episodes, although because RL exploration essentially "gives up" if an object isn't nearby, it can take lots of steps in many situations. In the real world, we saw similar behavior - sometimes, the RL policies would not explore enough to be able to find a goal at all.

| Nav. | Manip. | Perception | FindObj | Gaze | Pick | FindRec | Place | Place terminates |
|------|--------|------------|---------|------|------|---------|-------|------------------|
| Heuristic | Heuristic | Ground Truth | 100.0 | - | 65.1 | 65.1 | 62.1 | 62.1 |
| Heuristic | RL | Ground Truth | 100.0 | 65.6 | 64.3 | 64.3 | 61.3 | 52.2 |
| RL | Heuristic | Ground Truth | 100.0 | - | 76.3 | 76.2 | 66.8 | 66.8 |
| RL | RL | Ground Truth | 100.0 | 77.0 | 74.5 | 74.5 | 65.1 | 60.6 |
| Heuristic | Heuristic | DETIC [27] | 100.0 | - | 34.7 | 34.7 | 31.1 | 31.1 |
| Heuristic | RL | DETIC [27] | 100.0 | 33.9 | 27.2 | 27.2 | 24.4 | 17.6 |
| RL | Heuristic | DETIC [27] | 100.0 | - | 32.9 | 32.7 | 24.2 | 24.2 |
| RL | RL | DETIC [27] | 100.0 | 34.7 | 24.9 | 24.8 | 18.1 | 15.3 |

Table 6: We report the percentage of times each skill gets invoked for each of the different baselines. The last column gives the percentage of times the agent finishes executing all skills.

| Nav. | Manip. | Perception | FindObj Success. SC,UI | UC,UI | All | PickObj Success. SC,UI | UC,UI | Total | FindRec Success SC,UI | UC,UI | All | Overall Success SC,UI | UC,UI | All |
|------|--------|------------|------|------|-----|------|------|-------|------|------|-----|------|------|-----|
| Heuristic | Heuristic | Ground Truth | 50.9 | 53.2 | 54.1 | 46.4 | 47.2 | 48.5 | 27.5 | 30.0 | 31.5 | 4.1 | 5.2 | 5.1 |
| Heuristic | RL | Ground Truth | 54.9 | 58.2 | 56.5 | 48.6 | 55.1 | 51.5 | 39.0 | 37.3 | 42.3 | 14.5 | 12.0 | 13.2 |
| RL | Heuristic | Ground Truth | 67.1 | 64.1 | 65.4 | 55.0 | 51.0 | 54.8 | 44.8 | 44.8 | 43.7 | 6.2 | 7.8 | 7.3 |
| RL | RL | Ground Truth | 68.4 | 65.8 | 66.6 | 63.7 | 57.6 | 61.1 | 54.7 | 49.0 | 50.9 | 15.7 | 14.4 | 14.8 |
| Heuristic | Heuristic | DETIC [27] | 22.2 | 21.1 | 28.7 | 12.5 | 10.5 | 15.2 | 3.2 | 3.3 | 5.3 | 0.9 | 0.7 | 0.4 |
| Heuristic | RL | DETIC [27] | 22.4 | 22.2 | 29.4 | 11.9 | 11.8 | 13.2 | 5.1 | 3.5 | 5.8 | 0.3 | 1.4 | 0.5 |
| RL | Heuristic | DETIC [27] | 18.7 | 23.0 | 21.9 | 9.9 | 11.8 | 11.5 | 5.8 | 5.3 | 6.0 | 0.3 | 0.0 | 0.6 |
| RL | RL | DETIC [27] | 21.5 | 20.7 | 21.7 | 10.9 | 11.0 | 10.2 | 6.9 | 6.2 | 6.2 | 1.0 | 0.7 | 0.4 |

Table 7: Performance breakdown by seen and unseen categories, and compared to overall performance. In our baselines, we relied heavily on a pretrained object detector for generalization, so we don't see a dramatic difference in performance between seen and unseen objects.

Next, we observe that the Gaze and Place policies, which were trained with ground truth perception, take significantly longer to terminate with DETIC perception.

Finally, in Table 6, we look at the percentage of times the agent attempts each of the different skills. We find that the RL trained FindObj skill terminates more often than the heuristic FindObj skill and episodes terminate less frequently with DETIC perception when compared to GT perception.

## G.2 Performance on Seen vs. Unseen Object Categories

Table 7 shows results broken down by seen vs. unseen instances, and seen vs. unseen categories. Specifically we look at these two pools of objects from the validation set:

- **SC,UI:** Seen category, unseen instance. An object of a class that appeared in the training data (e.g., "cup"), but not a specific "cup" that appeared in the training data.

- **UC,UI:** Unseen instance of an unseen category; an object of a type that did not appear in the training data at all.

In general, because we are relying on DETIC and not training our own semantic perception for this baseline, we do not see a large difference between the two categories of object.

### G.2.1 Example DETIC predictions

In Table 5, we observe that the Gaze policy takes a significantly longer time to terminate with DETIC [27] perception. The gaze policy (see Fig. 19) tries to center the agent on the object of interest by allowing the agent to move its base and camera tilt. For this, it relies on DETIC's ability to detect novel objects. Now, we visualize DETIC segmentations of agent's egocentric observations by placing agent at the points where the Gaze skill is expected to start: the `object`'s viewpoints. We observe that while DETIC succeeds in a few cases, it fails at consistently detecting the objects in the egocentric frame.

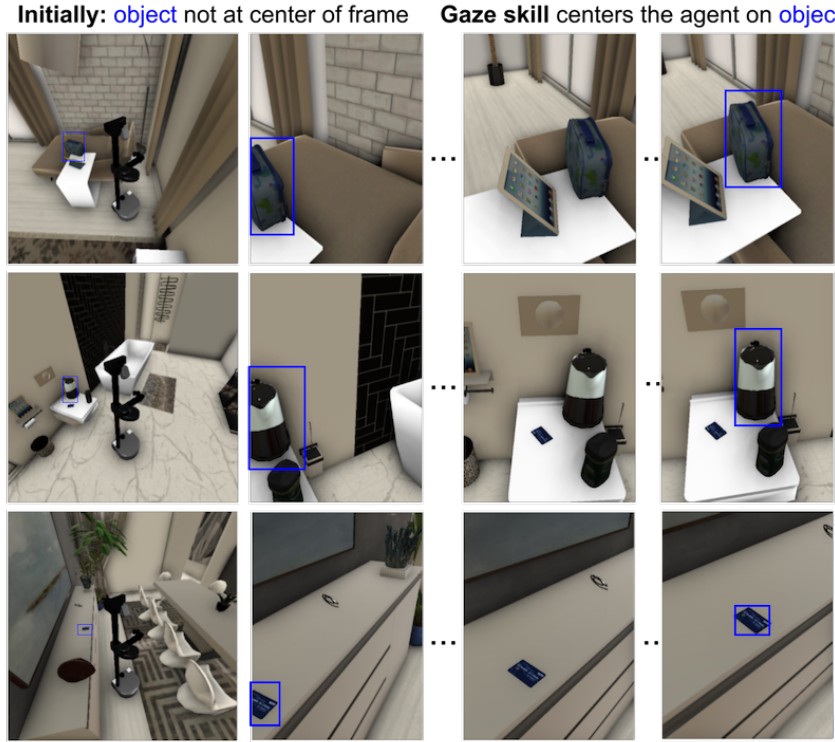

Figure 19: RL Gaze skill in action: The agent is allowed to move its base and change its camera tilt to get closer to `object` and bring `object` at the center of its camera frame

| Name | Mobile | Human Sized | Safe | Commercially Available | Manipulation DOF | Approximate Cost |
|---|---|---|---|---|---|---|
| Boston Dynamics Spot | ✔ | ✖ | ✖ | ✔ | 7 | $200,000 |
| Franka Emika Panda | ✖ | ✖ | ✓ | ✔ | 7 | $30,000 |
| Locobot | ✔ | ✖ | ✔ | ✖ | 5 | $5,000 |
| Fetch | ✔ | ✔ | ✓ | ✖ | 7 | $100,000 |
| Hello Robot Stretch | ✔ | ✔ | ✔ | ✔ | 4 | $19,000 |
| **Stretch with DexWrist** | ✔ | ✔ | ✔ | ✔ | 6 | $25,000 |

Table 8: Notes on platform selection. We chose the **Stretch with DexWrist** as a good compromise between manipulation, navigation, and cost, while being human-safe and approximately human-sized.

# H  Hardware Setup

Here, we will discuss choices related to the real-world hardware setup in extra detail along with information about the tools that we use for the visualization on the robot. This appendix contains notes on how to set up the robotics stack in the real world, useful tools that we contribute, and some best practices for development. Setting up mobile robots is hard, and one of the main goals of the HomeRobot project is to make it both easy and somewhat affordable for researchers.

## H.1  Hardware Choice

We describe some options for commercially available robotics hardware in Tab. 8. While the Franka Emika Panda is not a mobile robot, we include it here because it's a very commonly used platform in both industrial research labs and at universities, making its price a fair comparison point for what is reasonable.

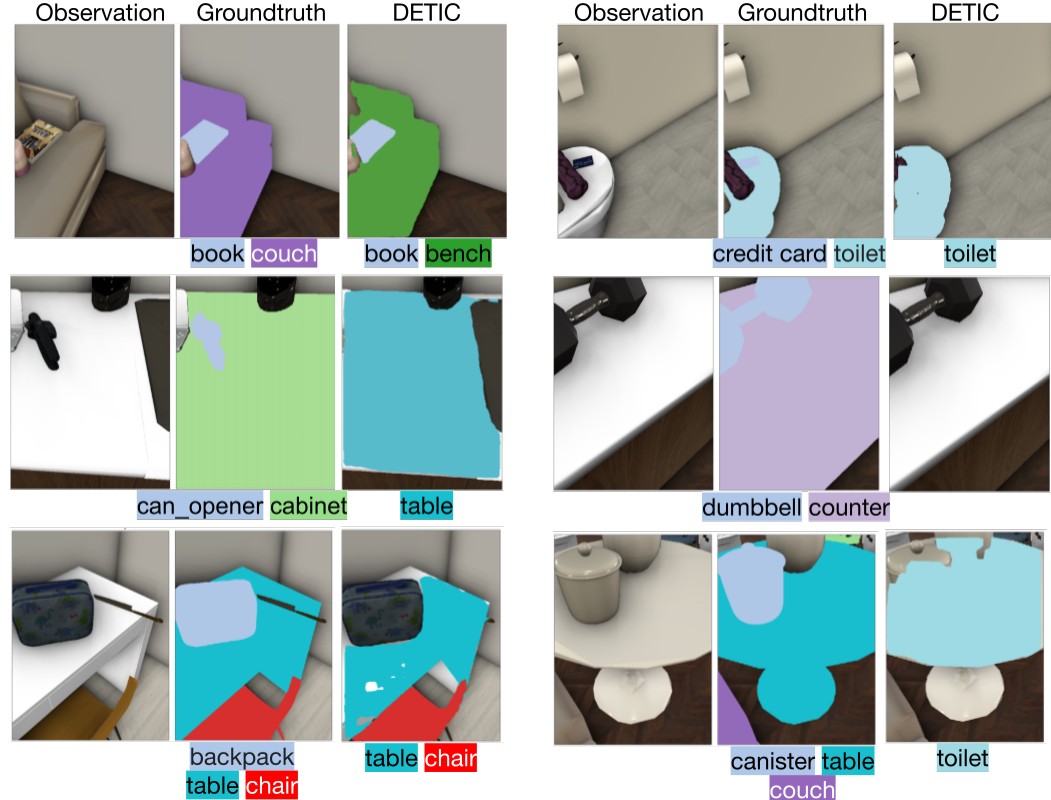

Figure 20: Visualization of groundtruth and DETIC [27] segmentation masks for agent's egocentric RGB observations. Note that we use a DETIC vocabulary consisting of the fixed list of receptacle categories and target `objectname`. We observed that DETIC often fails to accurately detect all the objects present in the given frame.

## H.2   Robot Setup

One challenge with low-cost mobile robots is how we can run GPU- and compute-intensive models to evaluate modern AI methods on them. The Stretch, like many similar robots, does not have onboard GPU, and will always have more limited compute than is available on a similar workstation.

As described in Sec. 4, we address this with a simple network configuration shown in Fig. 21. There are three components:

1. The **desktop** running code – in our case, the `eval_episode.py` script from HomeRobot – which connects to a remote mobile manipulator.

2. The dedicated **router** – an off-the-shelf consumer router, such as a Netgear Nighthawk router. This should ideally be dedicated for your robot and desktop setup to ensure good performance.

3. The mobile robot itself: a Stretch with DexWrist, as described above.

After the robot is configured, then you just need to run one script, a ROS launch file, as described in the HomeRobot startup instructions, which can be done over SSH. Then, with a properly configured robot and router, you can visualize information on the desktop side, showing the robot's position, map from SLAM, and cameras. On the robot side, the only necessary command is:

```
roslaunch home_robot_hw startup_stretch_hector_slam.launch
```

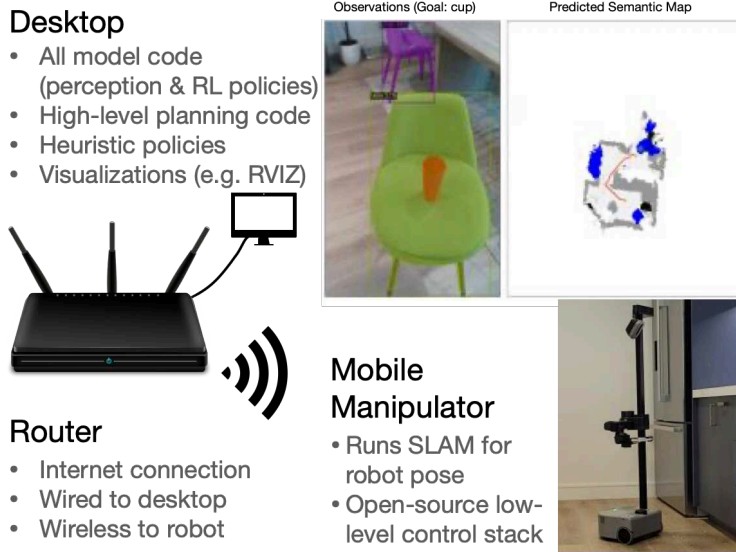

Figure 21: Network setup diagram for HomeRobot. We can run visualizations on a GPU-enabled workstation while running only the necessary code on a robot for low-level control and SLAM.

**Checking network performance.** We describe the visualization tools available briefly in the next section, but to check that the setup is working properly, you can start `rviz` and wave your hand in front of the robot – you should see minimal latency when waving a hand in front of the camera.

**Timing between the robot and the remote workstation.** We use ROS [113] as our communications layer, and to implement low-level control on the robot. This also provides network communication. However, due to potential latency between the robot and the desktop, we also need to make sure that observations are up to date.

We set up the robot to block after executing most navigation motions, in order to make this process simpler until there is an *up to date* image observation from the robot side. This means that timing between the robot and the workstation is extremely important: if we do not have up-to-date timing, we might have SLAM poses and depth measurements that do not match, which will lead to worse performance.

We solved this by having a clock on the robot side publish its time over ROS, and configuring all systems to use this ROS master clock instead of system time. This prevents the user from having to worry about Linux time synchronization protocols like NTP when setting up the robot for the first time.

### H.3 Visualizing The Robot

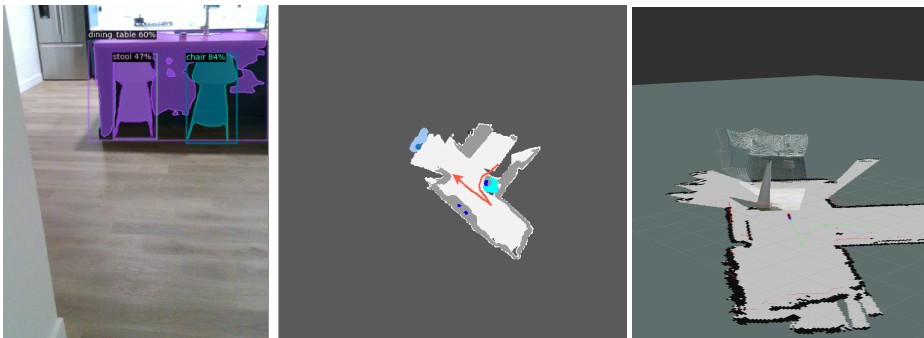

Figure 22: Exploring a real-world apartment during testing. The robot uses Detic [27] to perceive the world and update a 2D map (center) which captures where it's seen relevant classes, and which obstacles exist; detections aren't always reliable, especially given a large and changing vocabulary of objects that we care about. In the HomeRobot stack, we provide a variety of tools for visualizing and implementing policies, including integration of RVIZ (right).

We use RVIZ, a part of ROS, to visualize results and progress. Fig. 22 shows three different outputs from our system: on the far left, an image from the test environment being processed by Detic; in the center, a top-down map generated by the navigation planner described in Sec. E.2; and on the right, an image from RVIZ with the point cloud from the robot's head camera registered against the 2D lidar map created by Hector SLAM.

One advantage of the HomeRobot stack is that it is designed to work with existing debugging tools - especially ROS [113]. ROS is a widely-used framework for robotics software development that comes with a lot of online resources, official support from Hello Robot, and a rich and thriving open-source community with wide industry backing.

### H.4 Using The Stretch: Navigation vs. Position Mode

We leave API documentation to the HomeRobot code base, but want to note one other complexity when using the robot. Stretch's manipulator arm is pointed to the right of its direction of motion, which means that it cannot both look where it is going and manipulate objects at once. This allows the robot to be lower cost and fit the human profile - more information on the robot's design is available in other work [22].

However, it's something important to consider when trying to control Stretch to perform various tasks. We use Stretch in one of two modes:

- **Navigation mode:** the robot's camera is facing forward; we use reactive low-level control for navigation; the robot can rotate in place, roll backward, and will reactively track goals sent from the desktop.
- **Manipulation mode:** the robot's camera is facing towards its arm; we do not use reactive low-level control for navigation and do not rotate the base. Instead, we treat the robot's base as an extra, lateral degree of freedom for manipulation.

This is especially relevant when grasping or placing; it means that, for our heuristic policies, the robot transitions into manipulation mode after moving close enough to the goal, and may track slightly to the left or the right, in order to act as if it had a full 6dof manipulator.

All in all, these changes make the low-cost robot more capable and easier to use for a variety of tasks [12, 24, 25].

