# OpenReview forum: "HomeRobot: Open-Vocabulary Mobile Manipulation"
_robot-learning.org/CoRL/2023/Conference — CoRL 2023 Poster_

### Official Review · Reviewer_Jmt1 · 2023-07-18

**Confidence:** 3
**Originality:** Good
**Technical Quality:** Good
**Clarity Of Presentation:** Good
**Impact:** 3

**Recommendation:**

Weak Accept: I recommend accepting the paper, but will not argue for my recommendation if the majority of other reviewers have a different opinion.

**Review:**

The authors propose and study an important problem setting. I agree that OVMM is a useful measure of how the field has progressed toward its longstanding goal of general-purpose home robot assistants. Running the benchmark yearly to assess progress as the authors suggest can be very valuable to the community and of high impact, as it can highlight which components of OVMM require the most attention. The paper has a thorough appendix and useful videos for intuition. Table I is a helpful comparison to prior benchmarks and OVMM has an impressive amount of object instances compared to prior work.

However, the paper has several weaknesses listed below.
- The paper claims that OVMM is the “first reproducible mobile-manipulation benchmark for the real world,” and that existing efforts for reproducibility in robotics fall short due to lack of standardized hardware across labs. However, it is not clear how the proposed benchmark constitutes reproducible robotics. Existing initiatives include low-cost open-source hardware, large offline datasets, and shared remote hardware testbeds such as the Real Robot Challenge (https://arxiv.org/abs/2109.10957). The authors suggest the reproducibility is a result of the open-source software, low-cost robot ($25,000), and shared API for simulation and real world experiments, but (1) prior works have these components and (2) this does not solve the reproducibility issue in robotics: even if everyone adopted this hardware, it would not be possible to standardize the environment (lighting, friction, etc.). Furthermore, the fact that OVMM supports arbitrary objects *reduces* standardization rather than the suggested opposite. I would suggest either reducing the emphasis on reproducibility or providing further justification why OVMM is fundamentally different from previous attempts in this regard.
- The demonstration video showcases capabilities that do not appear to be state-of-the-art. The system is extremely slow even with the 4x speedup applied to the video, and the task is a simple pick-and-place action coupled with navigation. While HomeRobot may have a continuous action space it does not appear to be continuous control, as the robot strings together a sequence of short movements rather than a continuous one. Demos from projects such as SayCan (https://say-can.github.io/) seem to be far more proficient at the same task, with much more flexible natural language queries. Pick-and-place is also a significant limitation that should be added to the limitations section; most home tasks cannot be accomplished with a single pick-and-place (e.g., cooking, cleaning, laundry).
- HomeRobot code is not included in the supplement and is said to be released at the time of final submission, despite being a major claimed contribution of the paper. The existence of the robotics stack is a major differentiator in Table I, but it is not described with much detail in the main text.
- In general the use of the terms HomeRobot and OVMM seem to change throughout the text and overlap at times. For example, HomeRobot is defined in sentence 1 of the abstract as an affordable compliant robot for the home, and then later defined differently in the introduction as a software stack. The benchmark is sometimes called OVMM, HomeRobot OVMM, HomeRobot+OVMM. The contributions are also not the most clear. It would be ideal to clarify terminology throughout.

Also, there is existing work in scene graphs for multi-room environments that may be useful to include in related work, for example Hierarchical Mechanical Search (https://arxiv.org/abs/2012.04060) and SayPlan (https://arxiv.org/abs/2307.06135; looks like this was released after the CoRL submission deadline).

**Quality Of The Limitations Section:**

Additional details required

**Questions For Rebuttal:**

Please address the concerns raised in the weaknesses section above.

**Robotics Focus:**

Sufficient demonstration on hardware

**Summary Of Paper:**

This paper presents the Open-Vocabulary Mobile Manipulation (OVMM) benchmark. The authors define OVMM as the task of picking an arbitrary object in an environment and placing it at an arbitrary goal location, where the start and goal locations can be spread out over large multi-room areas and the objects do not belong to a predefined set. The purpose of OVMM is to measure progress in the field toward general purpose home robot assistants. The paper also presents the HomeRobot software stack for conducting OVMM experiments in simulation and real world environments. Reinforcement learning and heuristic approaches for OVMM are implemented in HomeRobot and compared in experiments.

**Summary Of Recommendation:**

**Post-Rebuttal Update**: I am upgrading to Weak Accept due to the authors' satisfactory rebuttal.

I recommend a Weak Reject due to the list of weaknesses identified above. While the OVMM problem setting is interesting, the paper's contributions over prior work are not clear to me. I may reconsider if the authors are able to address my concerns in the revision period, or the other reviewers have very different assessments of the work.

---

### Official Review · Reviewer_CNKn · 2023-07-19

**Confidence:** 4
**Originality:** Good
**Technical Quality:** Fair
**Clarity Of Presentation:** Good
**Impact:** 3

**Recommendation:**

Weak Accept: I recommend accepting the paper, but will not argue for my recommendation if the majority of other reviewers have a different opinion.

**Review:**

Strengths:
- I agree that it is the time that to have a large scale benchmark to standardize current robotic tasks both in the simulation and real-world environment. And the motivation of this work is quite clear and solid.
- The paper is organized and the definition of the OVMM task is clearly illustrated. The accompanied video and appendix provides some more implementation details about this work.

Weaknesses:
- The OVMM task is defined as "picking any object in any unseen environment, and placing it in a commanded location". Although the definition of the task is clear, the setup of this task demonstrated in the paper is actually a little simple. Only the seen and unseen objects are considered. However, in real-word environment, it is likely that there might be more complicated circumstances such as obstacles, impossible to be directly picked up, etc. I have some concerns about the adaptation ability of the proposed benchmark to more complex tasks. Also the benchmark does not support multiple agents.
- An heuristic and RL approaches are provided as baselines. And the ground truth and DETIC segmentation results are also used respectively. I wonder if there are some other stronger RL methods or 3D semantic information could also be included in the benchmark. With only two baselines on a single task, this benchmark seems a little weak and not promising enough.
- For the Sim2Real process, is there any sim2real training mechanism or directly transfer the trained model in the simulation to the real-world environment.
- Some minor problems:
  - In Fig.1, it is hard for me to distinguish the real-world from the simulation environment as indicated in the Line 57
  - Table 1 only lists some comparison aspects among different works, while no summarization or conclusion is given. It will be useful to have an analysis on these aspects.  Especially, I don't quite get the idea of what the robotics stack mean in the table.

**Quality Of The Limitations Section:**

Additional details required

**Questions For Rebuttal:**

As stated in the Weaknesses above.

**Robotics Focus:**

Sufficient demonstration on hardware

**Summary Of Paper:**

In this paper, a benchmark for the Open-Vocabulary Mobile Manipulation (OVMM) task is established. The benchmark consists of both a simulation component and a real-world component to implement the task of picking an object to a target location. This work takes a step forward to the standardization of some particular robotic tasks.

**Summary Of Recommendation:**

I like the idea to have a large scale benchmark to standardize current robotic tasks both in the simulation and real-world environment. However, the task setup in the proposed benchmark is a little simple and does not well align with real-world environment.

Thanks the authors for the rebuttal. The rebuttal explains some of my concerns, and now I am inclined to accept this paper.

---

### Official Review · Reviewer_kzRx · 2023-07-21

**Confidence:** 4
**Originality:** Good
**Technical Quality:** Very Good
**Clarity Of Presentation:** Good
**Impact:** 3

**Recommendation:**

Weak Accept: I recommend accepting the paper, but will not argue for my recommendation if the majority of other reviewers have a different opinion.

**Review:**

I like this paper. It's well written and related work is put in adequate context. Fair and comparable benchmarking is an important topic in robotics in general, and I applaude the authors' efforts to make a contribution in this direction. Large multi-modal models opened exciting possibilities for interfacing with robotic manipulation systems, so the presented line of work is of high practical relevance.

I do think that the authors ought to sharpen their contributions somewhat. The claimed sim-to-real transfer I find dubious, as their current simulation environment is basically limited to perception. However, in manipulation the most difficult (and critical for realism) simulation aspect is interaction/contact. Currently, the authors use a simple action (I assume rigid attachment to the end-effector) in place of physical interaction simulation. Therefore, I don't see how the proposed approach would stand up to existing solutions (e.g., the cited ManiSkill) which do provide a physics engine and thus ought to have a much better chance of sim-to-real transfer of interaction policies.

**Quality Of The Limitations Section:**

Additional details required

**Questions For Rebuttal:**

- Clarify contributions w.r.t. sim-to-real: without physics engine, I don't see how interaction policies could have a chance to transfer to hardware
- Maybe the title should reflect better the actual focus of the paper (benchmarking, software stack)
- Typo in line 210: should read "off-the-shelf policy"
- Tables 3 & 4 use different units (Table 3 uses percent, whereas Table 4 uses ratios) - that should be unified.

**Robotics Focus:**

Sufficient demonstration on hardware

**Summary Of Paper:**

This work contributes a benchmark and corresponding open-source software stack for the problem of open-vocabulary mobile manipulation using a relatively low-cost platform (Hello Robot Stretch) in simulation and on hardware. For benchmarking, the authors provide 60 hand-curated home scenes with  a large and diverse number of annotated target objects for manipulation. Furthermore, the paper provides two baseline solutions for the problem of target object transport (i.e., find object - pick object - locate receptacle - place object) based on planning and reinforcement learning respectively. These baselines achieve around 20% of overall success rate both in simulation as well as on hardware.

**Summary Of Recommendation:**

This paper shows no real theoretical contribution, but is valuable for practitioners. As the main topic is the provided software stack and benchmarking framework, this works' value will be determined by the provided software quality and uptake of the community which is hard to judge upfront. I do think that the presented work is technically sound and that a significant effort was put into it. As the topic of mobile manipulation for household robotics is of high practical interest, I do see real potential and would recommend the work for publication subject to the clarifications mentioned above.

---

### Official Review · Reviewer_uxxU · 2023-07-24

**Confidence:** 4
**Originality:** Good
**Technical Quality:** Good
**Clarity Of Presentation:** Very Good
**Impact:** 3

**Recommendation:**

Weak Accept: I recommend accepting the paper, but will not argue for my recommendation if the majority of other reviewers have a different opinion.

**Review:**

Strengths:
- Provides a benchmark and tools for evaluating open-vocabulary mobile manipulation in both simulation and on robot hardware (Hello Robot Stretch), which is timely and valuable to the community. The open source nature of HomeRobot library is also appreciated.
- The comparisons of the new benchmark against existing one (Table 1) is a useful review and provides context for what gaps are existing in robot simulators.
- The careful experimental design to test generalization across seen/unseen categories/instances is appreciated and a nice addition.

Weaknesses:
- Having the start receptacle always be available “to help agents know where to look for the object” seems like a pretty strong assumption, and seems counter to the OVMM vision since the goal is to handle unseen environments.
- Having all possible receptacle categories be seen during training seems like a strong assumption and seems like it may contribute to making this significantly simpler for navigation.
- Table 2 caption text is too close to the actual body text which makes it hard to read
- While the reviewer recognizes resolving all modeling simplifications of a simulator is not reasonable, the last of physical simulation of grasping makes training for the full OVMM in simulation hard, especially when cluttered scenes heavily impact affordances.
- The lack of motion planners is an important and standard component of a robotics stack that this is missing.


**Quality Of The Limitations Section:**

Limitations are addressed clearly

**Questions For Rebuttal:**

- What does “partial availability” mean for the manipulation column?
- What do the numbers in table 3 represent, task success rate (label that in caption if so). Do success rate numbers of a task like Pick depend on how often previous actions (FindObj) were successful?
- How is OMVMMAgent defined?
- In limitations, why are full natural language queries out-of-scope? What is meant by that?
- There are very little details on the Heuristic approach in section 4, what heuristics are used?
- In table 4, it seems the results for FindObj are significantly better than the simulation results, why is that the case, it seems very unintuitive.

[1] Chen, Boyuan, et al. "Open-vocabulary queryable scene representations for real world planning." 2023 IEEE International Conference on Robotics and Automation (ICRA). IEEE, 2023.


**Robotics Focus:**

Sufficient demonstration on hardware

**Summary Of Paper:**

This paper addresses the problem of Open-Vocabulary Mobile Manipulation OVMM). It proposes a definition for the key-task, provides an extensive benchmark for both simulation and robot hardware, and additionally includes both learned and heuristic baselines for evaluating how challenging the task is. Overall, the definition of the open-vocabulary mobile manipulation task does not feel particularly substantial since visual navigation and object retrieval has been well-studied within the robotics field recently with the advent of foundation models. The benchmark and simulation environment are welcomed, but the limiting assumptions prevent this benchmark from being as general as it first reads in the abstract. I weakly recommend this paper be accepted.

**Summary Of Recommendation:**

Overall, I recommend this paper be weakly accepted. The simulator and benchmark are the most substantial contribution, but without access to the robot hardware component (which many labs may not have), the value of HomeRobot simulator over ManiSkill-2 for example feels more minor. The paper is well written and clear, although the paper writing presents a grand vision for OVMM that the limiting assumptions prevents this paper from fully realizing. Since the main contribution would be useful for the community, I recommend this paper be accepted.

---

### Author Response · Authors · 2023-08-12
**Addressing common concerns**

We sincerely appreciate all reviewers’ time and efforts in reviewing our paper and for the constructive feedback. We are glad that the reviewers found our work "timely" (uxxU), "of high practical relevance" (kzRx) and its motivation "clear and solid" (CNKn). The reviewers appreciated our "careful experimental design" (uxxU) and the proposal of "an important problem setting" (CNKn). They also found our paper "well written" (kzRx), the appendix "thorough" and "useful" (CNKn, Jmt1), our literature review "helpful" and well "contextualized" (uxxU, kzRx, Jmt1), and the amount of object instances "impressive" (Jmt1).

We address the concerns shared by reviewers below.

- **Typos** (uxxU, kzRx, CNKn, Jmt1): We updated the paper to address all clarification questions and typos.
- **Physical simulation of manipulation** (uxxU, kzRx):
    - **Grasping** (uxxU, kzRx): We agree that leaving out physical simulation of grasping is a limitation of the current baseline. However, many existing platforms and systems - like the Boston Dynamics Spot - provide powerful grasping capabilities out-of-the-box. We implemented such a grasping policy for the Stretch as well. Simulating robust grasping that transfers from sim-to-real for arbitrary objects and environments is very challenging, which is why we decided to leave this to a separate component for now. Instead, we suggest relying on work like ContactGraspNet or numerous model-based or heuristic grasping approaches for real-world transfer. At the same time, we attempt to replace the current pick success criteria with a more realistic criteria that requires the agent to move its arm near the object without colliding with the scene. We added a baseline that performs top-down grasps resembling our real-world grasping policy, and resorts to side-ways grasps when the object is farther (see Fig. 5 and section B.1 in the updated draft).
    - **Placement** (kzRx): We do simulate physics during training and evaluation of the place skills. It is used to ensure that an object is stably placed on the receptacle by the agent and that the agent does not collide with objects in the scenes. Our RL baseline learns to move the robot’s arm and body to stably place objects and we see evidence of transfer of this skill in the real-world experiments.
- **Comparison to ManiSkill2** (uxxU, kzRx): Here we compare our work to the recently released simulation benchmark, ManiSkill2.
   - **ManiSkill2 has much smaller sets of scenes and objects.** We specifically identify dimensions in Table 1 where there is a notable difference between us and ManiSkill2. We note that we provide many more scenes (60 vs. only 1 per task), but more specifically we also provide a much larger visual diversity of objects for manipulation. Scenes in HomeRobot OVMM are much larger, with multiple rooms to explore (4+) instead of 1 per task, and therefore there is dramatically more visual diversity - ManiSkill uses only 25-30 variations for a single receptacle or object while keeping the rest of the scene the same, for example.
   - **ManiSkill2 focuses on low-level control for manipulation tasks**, which means that tasks are substantially smaller with a much shorter horizon. There are only four mobile manipulation tasks in ManiSkill2: opening a drawer, opening a door, pushing a chair, and moving a bucket. These tasks are analogous to our “place” skill in difficulty: they involve a relatively short horizon motion, where the goal is visible and there’s no exploration problem. While ManiSkill2 focuses on learning useful control policies, we focus on executing a very long horizon task, where goals are not visible or known beforehand, and it may take moving through multiple rooms and correctly detecting when an object (like the cabinets from ManiSkill2) is found. We believe there’s definitely room for synergy between the two problem sets; ManiSkill2 focuses much more on learning a variety of useful manipulation skills and incorporating this sort of complex dynamic interaction would be a great direction for future work.

- **Code release and uptake by community** (Jmt1, kzRx): We publicly released the codebase and proposed a challenge based on it at the time of submission. We additionally made the train and val datasets publicly available for participants and the research community to use. We are unable to add the link here because it would void anonymity. However, our code is already in use at roughly 20 different universities as of August’23. The codebase is large and has many dependencies; including it in the supplement was determined to be infeasible.

---

### Decision · Program_Chairs · 2023-08-30

**Decision:**

Accept (Poster)

**Comment:**

### Summary, Strengths and Weaknesses
This paper introduces the OVMM benchmark, which tackles the task of selecting any object from an unfamiliar setting and accurately placing it at a designated location. The benchmark encompasses both simulation and real-world elements, facilitating the assessment of diverse strategies. Reviewers agree that the mobile manipulation benchmark focusing on large-scale open-vocabulary conditions is important. The paper's structure is mostly clear and comprehensible.

The reviewers initially raised a few concerns on the definition of OVMM and limited physical interaction simulation, which is crucial in robotics. It is also regrettable that the authors failed to acknowledge prior contributions like RoboCup@Home, which has been organized to benchmark language-driven mobile manipulation using open and standard platforms (simulation, Toyota HSR, Pepper) since 2007.

The reviewers agree in their recommendation to accept the paper because most of the concerns have been addressed. I agree with their consensus.